# Systems genomics of salinity stress response in rice

Sonal Gupta[1], Simon Niels Groen[1,2,3], Maricris L Zaidem[1,4], Andres Godwin C Sajise[5], Irina Calic[6,7], Mignon Natividad[5], Kenneth McNally[5], Georgina V Vergara[5,8], Rahul Satija[1,9], Steven J Franks[6], Rakesh K Singh[5,10], Zoé Joly-Lopez[11]*, Michael D Purugganan[1]*

[1]Center for Genomics and Systems Biology, New York University, New York, United States; [2]Department of Nematology and Department of Botany & Plant Sciences, University of California, Riverside, Riverside, United States; [3]Center for Plant Cell Biology, Institute for Integrative Genome Biology, University of California, Riverside, Riverside, United States; [4]Department of Biology, University of Oxford, Oxford, United Kingdom; [5]International Rice Research Institute, Los Baños, Philippines; [6]Department of Biological Sciences, Fordham University, Bronx, United States; [7]Inari Agriculture Nv, Gent, Belgium; [8]Institute of Crop Science, University of the Philippines, Los Baños, Philippines; [9]New York Genome Center, New York, United States; [10]International Center for Biosaline Agriculture, Dubai, United Arab Emirates; [11]Département de Chimie, Université du Quebéc à Montréal, Montreal, Canada

*For correspondence:
joly-lopez.zoe@uqam.ca (ZJ-L);
mp132@nyu.edu (MDP)

## eLife Assessment

Working with a diverse panel of rice accessions grown in field conditions, this **valuable** study measures changes in transcript abundance, tests for patterns of selection on gene expression, and maps the genetic basic of variation in gene expression in normal and elevated salinity treatments. The manuscript provides **solid** evidence that mean gene expression levels are further from the optimum abundance for more genes under the elevated salinity treatment compared to normal treatment, and that a relatively small number of genes are hotspots that harbor genetic variants which affect broader genome-wide patterns of natural variation in gene expression under high salinity conditions. However, the design, clarity, and interpretation of several statistical analyses can be improved, some opportunities for integration among datasets and analyses could yet be realized, and genetic manipulation is required to confirm functional involvement of any specific genes in regulatory networks or organismal traits that confer adaptation to higher salinity conditions. The manuscript will be of interest to evolutionary biologists studying the genetics of complex traits and a resource for plant biologists studying mechanisms of abiotic stress tolerance.

**Abstract** Populations can adapt to stressful environments through changes in gene expression. However, the fitness effect of gene expression in mediating stress response and adaptation remains largely unexplored. Here, we use an integrative field dataset obtained from 780 plants of *Oryza sativa* ssp. *indica* (rice) grown in a field experiment under normal or moderate salt stress conditions to examine selection and evolution of gene expression variation under salinity stress conditions. We find that salinity stress induces increased selective pressure on gene expression. Further, we show that *trans*-eQTLs rather than *cis*-eQTLs are primarily associated with rice's gene expression under salinity stress, potentially via a few master-regulators. Importantly, and contrary to the expectations, we find that *cis-trans* reinforcement is more common than *cis-trans* compensation which may be reflective of rice diversification subsequent to domestication. We further identify genetic fixation

as the likely mechanism underlying this compensation/reinforcement. Additionally, we show that *cis*- and *trans*-eQTLs are under balancing and purifying selection, respectively, giving us insights into the evolutionary dynamics of gene expression variation. By examining genomic, transcriptomic, and phenotypic variation across a rice population, we gain insights into the molecular and genetic landscape underlying adaptive salinity stress responses, which is relevant for other crops and other stresses.

## Introduction

Plants face numerous stresses that reduce their growth and fitness, and they have a variety of adaptations to help them deal with environmental challenges (*Mareri et al., 2022*). For crop plants, stresses such as drought, heat, and salinity can greatly limit productivity and agricultural sustainability worldwide (*Fahad et al., 2017*; *Kopecká et al., 2023*); indeed, as a direct result of various abiotic stresses, an estimated ~50–70% of crop yields are lost (*Francini and Sebastiani, 2019*). While there is a wealth of information on the physiological responses of plants, including crops, to stress (*Zhang et al., 2022a*), we are still developing an understanding of the underlying genetic responses to stress and the complex interactions between stresses and environmental cues, networks of gene expression regulation, physiological responses and ultimately plant fitness (*Cramer et al., 2011*; *Siddiqui et al., 2021*).

Much of the research on the genetic basis of stress responses in crops has focused on identifying individual genes associated with specific responses and/or tolerance traits (*Cui et al., 2020*; *Jiang et al., 2019*; *Kumar and Wigge, 2010*; *Laohavisit et al., 2013*; *Yuan et al., 2014*). Stress responses, however, are complex traits that can be influenced by multiple genetic pathways with numerous interconnected genes. Expression levels of genes are controlled by regulatory elements and are often environmentally induced, suggesting a critical role of regulatory divergence in adaptive evolution (*De Clercq et al., 2021*; *Wilkins et al., 2016*). Furthermore, changes in gene expression can lead to changes in the transcript network correlational structure, reshaping regulatory pathways (*Lea et al., 2019*) and influencing the way in which gene expression variation can influence adaptation to stressful environments (*Signor and Nuzhdin, 2018*; *Wagner and Lynch, 2008*). As such, there is an increasing need to understand how evolutionary dynamics of gene expression and transcript abundance relate to the genetic underpinnings of stress responses and adaptation in crops (*Groen et al., 2020*; *Ruffley et al., 2023*; *Siddiqui et al., 2021*).

One of the most important stresses for many plants is salinity. Soil salinity causes osmotic imbalance between the plant and the soil, which impedes the uptake of water and other key nutrients (*Hakim et al., 2014*; *Munns, 2002*), possibly leading to acute ion toxicity (*Liang et al., 2018*). While some plants are considered halophytic and can thrive in saline environments, other plants are highly sensitive to salts and experience negative effects of salinity even at low concentrations (*Carillo et al., 2011*). *Oryza sativa* (Asian rice) is one such salt-sensitive crop, facing significant yield loss due to soil salinity (*Hussain et al., 2017*; *Melino and Tester, 2023*). Rice responds to salinity stress by adjusting physiological and biochemical processes involved in osmotic and ion homeostasis, nutritional balances and oxidative stress (*Castillo et al., 2007*; *Miller et al., 2010*; *Munns and Tester, 2008*; *Qin and Huang, 2020*; *Wang et al., 2012*). Although studies have identified multiple genes associated with these processes (for reviews, see *Ponce et al., 2021* and *Liu et al., 2022*), we still lack a systems-level understanding of how gene expression variation mediates responses to salinity stress and evolves at the molecular level.

Here, we use an integrative system genomics approach to comprehensively dissect the genome-wide molecular and phenotypic response to salinity stress in rice. This study builds on prior work by our group that examined selection on gene expression in rice in response to normal and dry conditions (*Calic et al., 2022*; *Groen et al., 2020*; *Groen et al., 2021*). Using genomic, transcriptomic, and phenotypic datasets obtained from 130 diverse accessions of rice subjected to moderate levels of salinity stress, we (i) explore the selection on gene expression variation under salinity stress, (ii) dissect the genetic architecture of gene expression variation under saline conditions, and (iii) identify genes, molecular pathways, and salt stress response traits as well as associated trade-offs in the saline environment. We demonstrate that salinity stress induces increased selective pressure on gene expression, and we identify variation in biological processes and physiological traits that is beneficial and

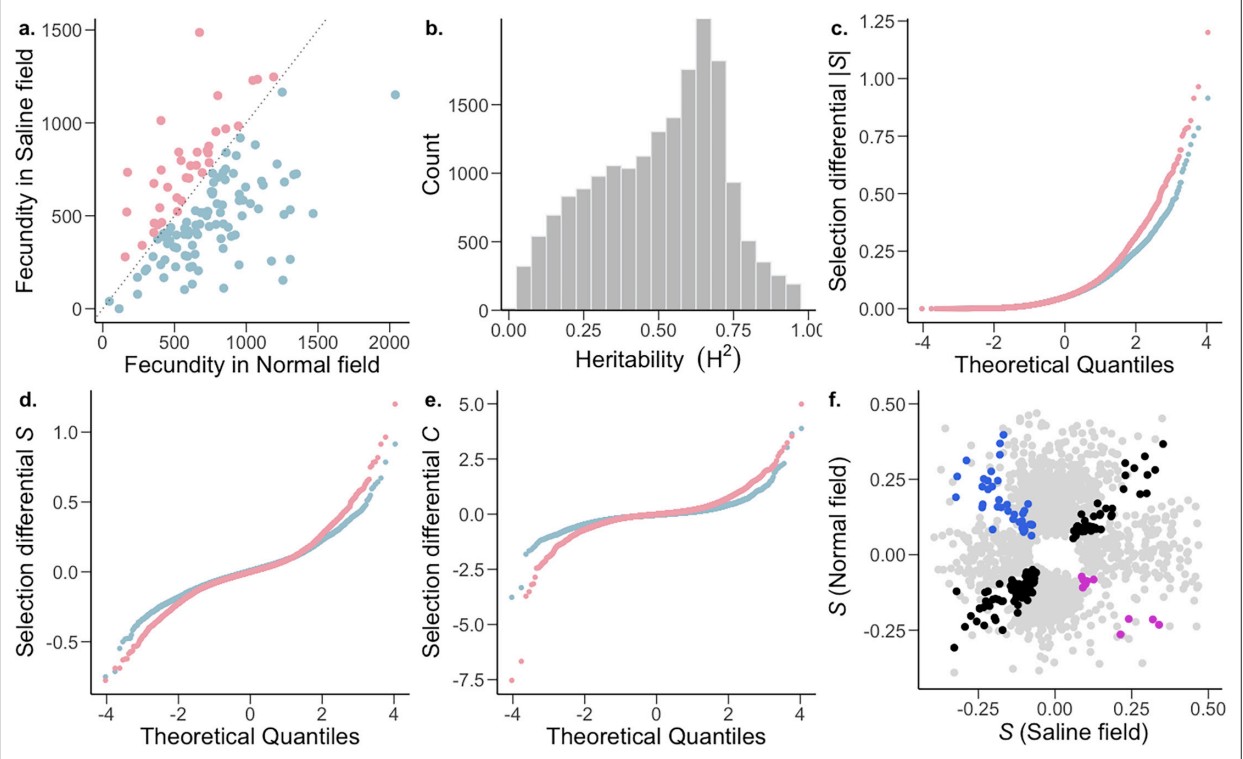

**Figure 1.** The strength and pattern of selection on heritable gene expression. (**a**) The *O. sativa* ssp. *indica* populations showed higher average fitness in the normal (blue) and saline (pink) field (two-tailed paired t-test p=1.658 x 10⁻⁸) and fitness further showed a significant effects of genotype (G) and environment (E); genotype ×environment (G×E) was not significant. Analysis of variance (ANOVA) [G and E (p<0.001), G×E (p=0.49)]; n=130 accessions. (**b**) Broad-sense heritability (H²) distribution of *Oryza sativa* spp. *indica* transcripts. Two-way ANOVA, genotype FDR-adjusted q<0.001, n=130 accessions. (**c–e**) The strength of linear selection |S|, linear selection differentials (S), and quadratic selection differentials (C) for genome-wide gene expression in normal (blue) and saline (pink) conditions. X-axes represent a theoretical quantile for normal distribution with mean = 0 and standard deviation = 1. (**f**) Conditionally neutral (light gray), and antagonistically pleiotropic transcripts (blue and magenta represent beneficial expression in normal and saline conditions, respectively). Black represents transcripts experiencing selection in the same direction in both environments (expression is beneficial or detrimental in both environments).

The online version of this article includes the following figure supplement(s) for figure 1:

**Figure supplement 1.** Pathway enrichment of the 51 antagonistically pleiotropic genes beneficial in normal conditions but detrimental in salinity stress conditions.

**Figure supplement 2.** Boxplot representation of eQTLs for the two photosynthesis-related AP (antagonistically pleiotropic) genes beneficial in normal conditions.

detrimental to plants in a saline environment, providing novel insights into the molecular landscape underlying an adaptive response to excess salt. We integrate these datasets with genomic sequence information to elucidate the genetic architecture and regulatory networks governing rice's response to salinity stress.

## Results

### Variation in transcript abundance

To understand the microevolutionary dynamics of gene expression variation under salinity stress, we first investigated variation in transcript abundance in 130 accessions of *O. sativa* ssp. *indica*. We conducted a field experiment in the dry season of 2017 (January-May) at the International Rice Research Institute in Los Baños, Laguna, Philippines. Three replicates of each accession were planted separately in a normal wet paddy field as well as a similar field in which plants were exposed to moderate salinity stress (salt levels maintained at 6 dSm⁻¹) maintained until maturity. Average fecundity was significantly lower in the saline field than in normal conditions (*Figure 1a*; two-tailed paired

t-test p=1.658 x 10⁻⁸) with most of genotypes having significantly lower fecundity in the saline field (n=94; one-sample proportion test p=3.639 x 10⁻⁷).

To examine gene expression in the field, mRNA levels were measured in leaf blades from 780 plants (130 accessions in triplicates for each environment) via 3'-end-biased mRNA sequencing (*Meyer et al., 2011*). Leaf blades were sampled 38 days after sowing (DAS), corresponding to 7 days of plants being exposed to stress in the saline field. Population variance in gene expression of 18,141 widely expressed transcripts was partitioned into genotype (G), environment (E), and genotype ×environment (G×E) effects using a two-way mixed analyses of variance (ANOVA) with environment as a fixed effect and genotype and genotype ×environment as random effects (*Supplementary file 1*). At a conservative false discovery rate (FDR) of 0.001, all but 3 transcripts displayed a significant genotype effect, indicating that expression levels of most transcripts are heritable. This effect of genotype is reflected in the broad-sense heritability ($H^2$) distribution of gene expression levels (*Figure 1b*), which had a median value $H^2$=0.53 (range of 0.012–0.987; *Supplementary file 1*). In addition to the high levels of heritability, 16,371 transcripts had a significant G×E term, indicating that for many transcripts, genotypes showed heritably different levels of expression in different environments. The significant G×E indicates genetic variation for plasticity, indicating that this plasticity can evolve. Although we see a widespread heritable plastic response in gene expression, we found no evidence of genotype-dependent plasticity in fitness (G×E for fitness p=0.49; *Figure 1a*), indicating that the G×E of transcripts does not translate to the complex trait of fitness. This could be due to a combination of factors, like gene interactions (pleiotropy and epistasis) leading to little to no effect on fitness, or environment specific genotype-dependent gene regulation (environment specific eQTLs). Furthermore, only a relatively small number of transcripts (254) showed significant variation due to E, indicating genotype-independent plasticity in gene expression for only a small fraction of genes.

## Selection on gene expression

To identify transcripts associated with high fitness in normal and saline environments, we measured the strength of selection on gene transcript levels. We did this using phenotypic selection analysis (*Lande and Arnold, 1983*), taking the total number of filled rice seeds/grains (total fecundity) as a proxy for fitness (*Groen et al., 2020*). We estimated the linear ($S$) and quadratic ($C$) selection differentials, which are estimates for directional (negative or positive $S$) selection, and stabilizing (negative $C$) or disruptive (positive $C$) selection. We calculated the raw ($S$ and $C$), variance-standardized ($S_s$ and $C_s$), and mean-standardized differentials ($S_m$ and $C_m$). We found that both mean and variance of transcript expression vary significantly between conditions (Mann-Whitney $U$-test, mean and standard deviation: p<2.2 x 10⁻¹⁶), which means that the interpretation of the strength of selection based on either of these differentials could be misleading (*Supplementary file 2*). To overcome this, and given that we were interested in selection on gene expression, which were all measured in the same units, we used the non-standardized raw selection differentials for all downstream analyses. We also report the mean- and variance-standardized selection coefficients (*Supplementary file 3*). From hereon, we

**Table 1.** Summary statistics of selection on gene expression.

|  |  | Control | Salt |
|---|---|---|---|
| Median |$S$| |  | 0.0501 | 0.0507 |
|  | # Transcripts | 9654 | 8885 |
| $S>0$ | Median $S$ | 0.053 | 0.051 |
|  | # Transcripts | 8415 | 9133 |
| $S<0$ | Median $S$ | −0.047 | −0.050 |
|  | # Transcripts | 7175 | 8713 |
| $C>0$ | Median $C$ | 0.077 | 0.115 |
|  | # Transcripts | 10894 | 9304 |
| $C<0$ | Median $C$ | −0.096 | −0.111 |

S and C represent the linear and quadratic selection differentials.

use positive and negative directional selection to represent the case where higher fitness is associated with increase and decrease in trait value, respectively. Stabilizing and disruptive selection similarly represent the situations where higher fitness is associated with average vs extreme value of the trait, respectively. In contrast, we limit the use of positive and purifying selection to their molecular evolution definition, that is to represent the case where increase in fitness is associated with derived and ancestral allele, respectively. Lastly, balancing selection refers to the circumstance where multiple alleles are maintained in the population which is associated with increased fitness.

Most transcripts were (nearly) neutral ($|S|<0.1$) in both the normal and saline environments (*Figure 1c*). Additionally, within the normal environment, there was a general trend towards stronger positive (9654 transcripts with $S>0$) compared to negative (8,415 transcripts with $S<0$) selection on gene expression, indicating that higher expression of transcribed genes is associated with greater fitness (*Table 1*, Mann-Whitney $U$-test, normal: p=3.95 x $10^{-15}$). In comparison, although the strength of positive directional selection was higher than that of negative directional selection in the saline field, higher expression of most of the transcribed genes is associated with lower fitness (*Table 1*, Mann-Whitney $U$-test, saline: p=0.0068). Although no transcripts cleared the Bonferroni correction threshold in saline conditions, under normal conditions 17 transcripts cleared the threshold, 13 of which were under positive directional selection (*Supplementary file 3*).

We also found that a high proportion of transcripts experienced stabilizing selection ($C<0$) in both normal and saline environments. This effect was pronounced under normal conditions, with the strength of stabilizing selection being stronger than that of disruptive selection (*Table 1*; Mann-Whitney $U$-test, normal: p<2.2 x $10^{-16}$). In contrast, within the saline environment, there was no detectable difference between the strength of stabilizing and disruptive selection (Mann-Whitney $U$-test, normal: p=0.514), due to a higher proportion of transcripts experiencing stronger disruptive selection. This indicates that in the saline environment, as compared to normal conditions, an increase in fitness is associated with extremes in transcript abundance.

Comparing the distribution of selection differentials between environmental conditions, we found that selection was stronger in the saline field compared to the normal wet paddy field (*Table 1*; *Figure 1d and e*). This was true for the overall strength of directional selection (Mann-Whitney $U$-test, p=0.012), as well as for negative directional (p=8.27 × $10^{-6}$), stabilizing (p<2.2 x $10^{-16}$) and disruptive (p<2.2 x $10^{-16}$) selection, but not positive directional selection (p=0.735).

## Gene expression trade-offs

Next we compared the proportions of genes showing an opposite direction of selection in across environments (antagonistic pleiotropy- AP) and those showing selection in only one environment (conditional neutrality- CN; *Anderson et al., 2011*). Since the detection of CN relies on p-value being significant in only one environment, compared to the detection of AP which relies on p-value being significant in both environments, this introduces a bias towards the detection of CN. To account for this inherent bias, we used a more stringent p-value cutoff to define CN (a transcript with p<0.025 in one environment and p>0.05 in the other environment) in comparison to AP (a transcript with p<0.05 in both environments and opposite directionality of $S$; *Groen et al., 2020*; *Siddiqui et al., 2021*). We found that 10.80% of the transcripts showed selection patterns consistent with CN, while only 0.28% of transcripts showed AP (*Figure 1f*; *Supplementary file 4*). The proportion of transcripts showing CN was greater than expected by chance and greater than the proportion showing AP two-tailed proportion z-test p<2.2 x $10^{-16}$. These results are consistent with a lack of trade-offs in which the expression of a gene is favored in one environment and disfavored in another.

Among the 51 AP transcripts, increased expression of 38 was beneficial only in normal conditions (higher expression associated with higher fitness in normal conditions, but lower fitness under saline conditions). Gene ontology (GO) term analyses of these 38 transcripts indicated that many of these genes were involved in photosynthesis and metabolic processes (*Figure 1—figure supplement 1*; *Supplementary file 4*). This is consistent with an observed reduction of photosynthesis in rice under salinity stress (*Radanielson et al., 2018*; *Tsai et al., 2019*).

We then wondered whether any difference underlie gene regulation of these AP transcripts relative to the non-AP transcripts, potentially indicating the genetic basis for this trade-off. We identified single nucleotide polymorphisms (SNPs) associated with transcript expression levels in each environment separately using whole-genome polymorphism data (expression quantitative trait loci

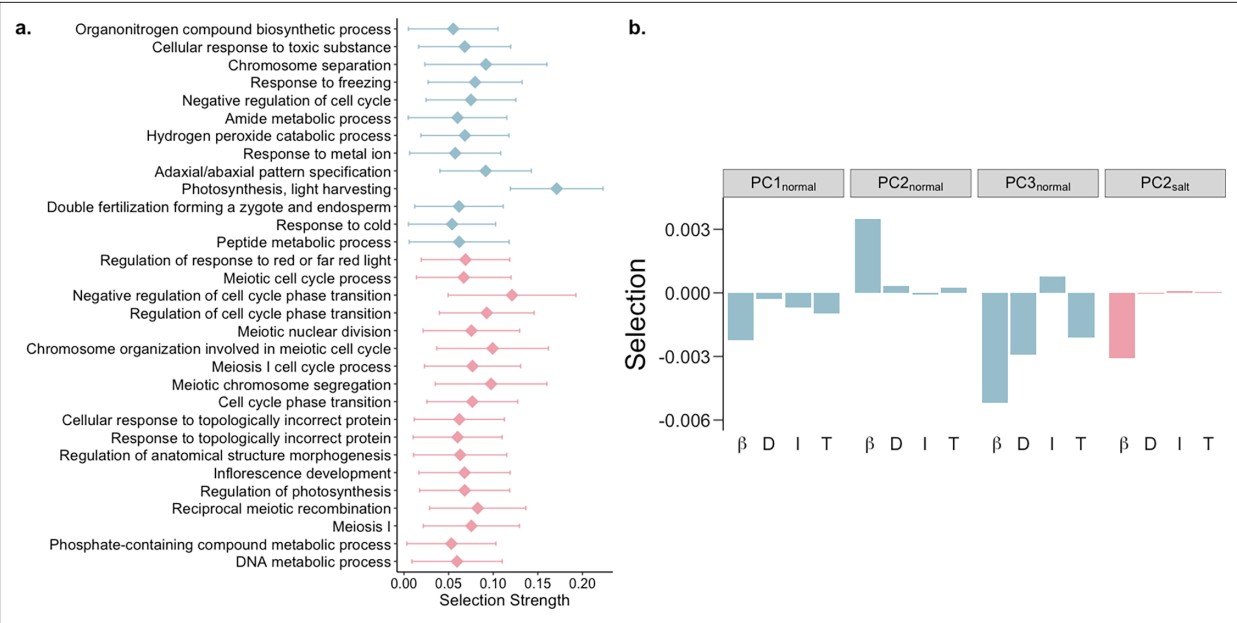

**Figure 2.** Biological processes and pathways with differential responses to selection under saline conditions. (**a**) GO biological processes under stronger selection in normal (blue) and saline conditions (pink). Error bars represent 95% confidence intervals. around the median. (**b**) Linear selection gradients (β), along with direct (D), indirect (I) and total (T) responses to selection on suites of transcripts in normal (blue) and saline conditions (pink).

The online version of this article includes the following figure supplement(s) for figure 2:

**Figure supplement 1.** Enrichment of the suite of transcripts (*1% tails of the distributions of transcripts' loading values on principal components*) with significant selection gradients in both normal and salinity stress conditions.

[eQTL] analyses; see below). We found SNPs regulating expression of two photosynthesis related AP transcripts (*PSAN* and *CRR7*) only in the normal environment (*Figure 1—figure supplement 2*). We further looked at the 13 AP transcripts that were beneficial only in the saline environment (*Supplementary file 4*). Although there was no significant GO enrichment for these transcripts, among them we identified a cyclophilin-encoding transcript (*OsCYP2*), which has been shown to confer salt tolerance in rice (*Lee et al., 2015*; *Roy et al., 2022*; *Ruan et al., 2011*). However, no SNP was associated with the expression of this gene in either normal or saline conditions.

## Biological processes under selection

To investigate the broader biological processes associated with differential selection (strong directional selection in only one environment), we ranked all GO biological processes by their median directional selection strength in each environment and identified the processes with significantly stronger selection relative to their respective environment. We identified 13 and 18 processes that were under strong differential selection in normal and saline conditions, respectively (*Supplementary file 5*; *Figure 2a*). Processes primarily involved in various aspects of growth and defense were under stronger selection in normal conditions, whereas processes associated with regulation of flowering, cell cycle control and reproduction showed stronger selection under saline conditions (*Figure 2a*). This provides insight into the specific biological processes related to changes in flowering time and reduced yield, both of which have been associated with salinity stress in multiple species (*Chang et al., 2019*; *Li et al., 2007*; *Zandt and Mopper, 2002*; *Zhang et al., 2022b*). Furthermore, studies have found that the osmotic stress induced by salinity causes a reduction in the cyclin-dependent kinases (CDKs) responsible for cell-cycle transitions (G1/S and G2/M; *Ma et al., 2015*; *Schuppler et al., 1998*; *West et al., 2004*). Aligned with this, our study also supports the notion that salinity stress affects cell cycle regulation, and leads to reduced growth and reproduction.

Gene expression usually operates within the context of robust gene interaction/regulatory networks (*Amiri et al., 2018*; *Israel et al., 2016*; *Ko and Brandizzi, 2020*). Selection acting on these interacting genes is one of leading causes of indirect selection, which can constrain the response of a population to selection on gene expression (*Agrawal and Whitlock, 2010*; *Groen et al., 2020*; *Kondrashov and*

*Houle, 1994*). To examine this phenomenon, we identified suites of correlated transcripts using principal component (PC) analysis on genome-wide gene expression levels. We then estimated the linear (β) and quadratic selection (γ) gradients for PCs explaining over 0.5% of variance in each environment (*Supplementary file 6*). Although quadratic selection was generally weak, linear selection on some PCs showed significant directional selection (*Supplementary file 7*).

Using the breeder's equation (*Falconer and Mackay, 1996*), we predicted the response to selection on these PCs to examine the constraints on a population's microevolutionary response to selection on gene expression. We found that over half of PCs (7 of 11 and 7 of 12 PCs in normal and saline conditions, respectively) displayed opposite signs of direct and indirect responses to selection (*Supplementary file 7*). This finding contrasts with prior work in rice under dry and wet conditions that found a lack of constraint in response to selection for most traits except seed size because direct and indirect responses to selection were largely similar (*Calic et al., 2022*), indicating that different types of stress may either constrain or facilitate the response to selection.

We found PCs enriched for metabolic pathways and biosynthesis of phenylpropanoids and secondary metabolites to be under selection in the normal environment (*Figure 2a*; *Figure 2—figure supplement 1*; *Supplementary file 7*). Different transcripts involved in these pathways were under positive and negative directional selection which may act to keep these pathways in a steady-state. Interestingly, circadian rhythm was found to be under positive directional selection, with an overall positive response to selection, in the saline environment (*Figure 2a*; *Figure 2—figure supplement 1*; *Supplementary file 7*). This is in alignment with the role of circadian clock genes in conferring salt tolerance (*Kim et al., 2013*; *Wei et al., 2021*; *Xu et al., 2022*), and indicates a tentative increase in expression of circadian clock genes with continuous exposure to soil salinity.

## Salinity stress induces decoherence

Gene expression levels are generally correlated, but these correlations can be perturbed by environmental stresses, a phenomenon that is termed decoherence (*Lea et al., 2019*). Such decoherence has been demonstrated in humans and primates (*Lea et al., 2019*; *Pu et al., 2022*; *Watowich et al., 2022*), but little is known about how stress alters the correlation structure of specific transcripts pairs, and functional groups of transcripts, in plants. To examine decoherence in rice, we utilized the recently developed CILP (Correlation by Individual Level Product) method (*Lea et al., 2019*), which detects the systematic loss of correlation in gene expression among individuals. Since CILP calculates product correlations for all possible pairs of genes, we used transcripts with selection strengths greater than 0.1 ($|S|>0.1$) in at least one environment with expression greater than 0 in at least 50% of individuals (2051 transcripts; *Supplementary file 8*) to reduce data dimensionality.

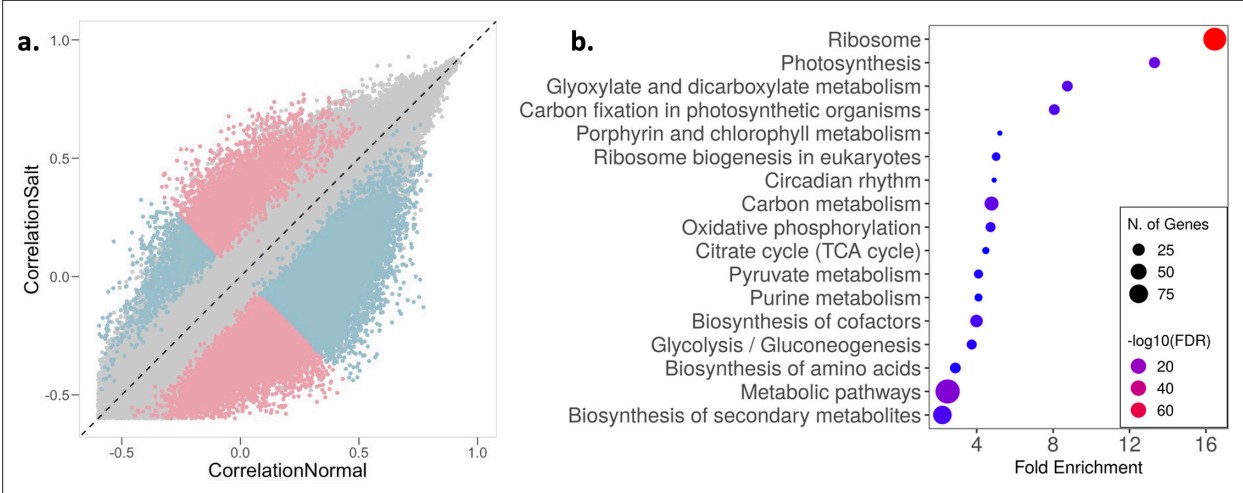

**Figure 3.** Salinity stress induces regulatory decoherence. (**a**) Pearson correlation coefficients between pairs of transcripts ($|S|>0.1$ and expression greater than 0 in at least 50% individuals) in normal (x-axis) and saline conditions (y-axis). Pink and blue represent pairs with correlation stronger in saline and normal conditions, respectively; gray represents correlation that is not significantly different between conditions. (**b**) Enrichment of transcripts with significant pairs greater than the median (median significant pair per transcript = 12, n=853) involved in regulatory decoherence post salt exposure.

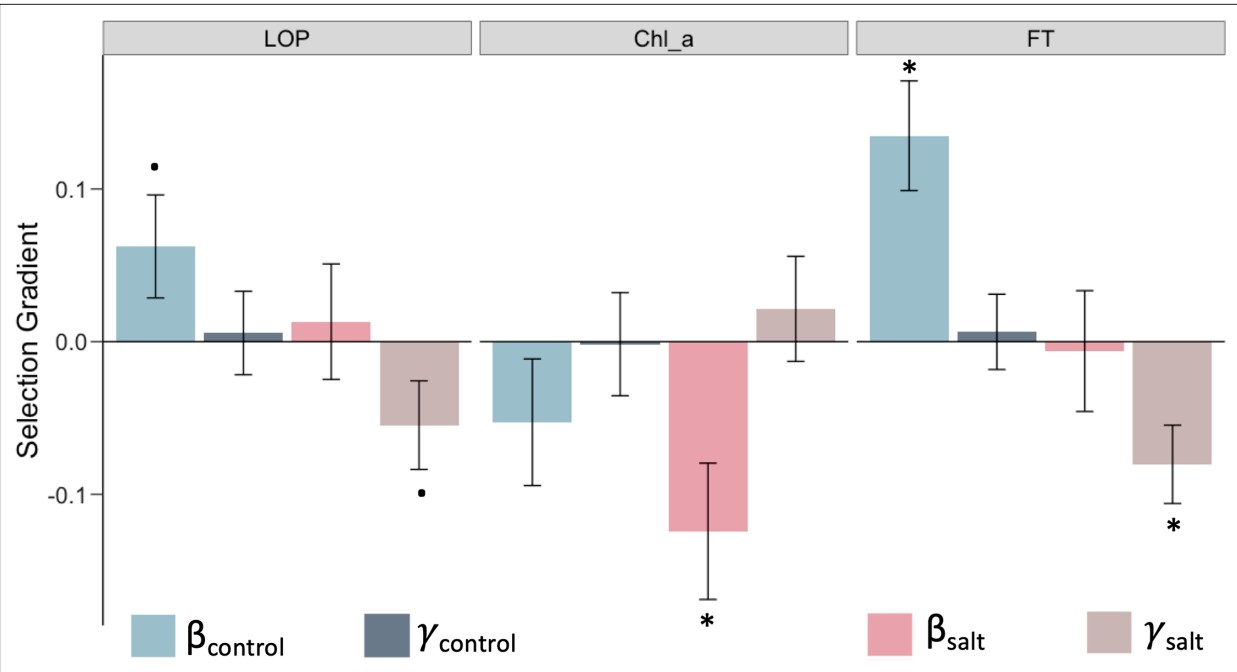

**Figure 4.** Traits with different selection profiles under salt stress. Linear ($\beta$) and quadratic ($\gamma$) selection gradients on the traits LOP (leaf osmotic potential), Chl_a (chlorophyll a content), and FT (flowering time). Error bars represent mean ± SE ($n_{normal}$: 384; $n_{salt}$:365); dots and asterisks indicate significance of selection-gradient at two-sided unadjusted $p<0.1$ and $p<0.05$.

The online version of this article includes the following figure supplement(s) for figure 4:

**Figure supplement 1.** Correlation among functional traits in normal conditions.

**Figure supplement 2.** Distribution of flowering time (representing days to when 50% of plants in a plot flowered) in normal and salinity stress conditions.

We found that the correlation structure for gene expression was broadly similar across the 2.10 M unique transcript pairs, but >29,000 transcript pairs (involving 1742 unique transcripts) showed significantly different correlations between normal and saline conditions (FDR <5%) (*Figure 3a*; *Supplementary file 8*). This change in correlation structure indicates possible divergence in gene interactions between environments, which may presage a restructuring of gene networks. GO term enrichment analysis of the transcripts that show decoherence with significant pairs greater than the median (median significant pair per transcript = 12, n=853) highlighted important pathways related to plant responses and potentially tolerance to excess salt (*Figure 3b*). For instance, circadian rhythm genes like *OsPRR37* and GIGANTA (*GI*), which have been shown to confer salt tolerance in rice and *Arabidopsis* (*Kim et al., 2013*; *Wei et al., 2021*), differed significantly in their interactions between the two environments. Similarly, to manage the energy requirements of salt stress, the relative abundance of metabolites involved in energy producing pathways like glycolysis, tricarboxylic acid (TCA) cycle and various other metabolic processes have been shown to be altered as an early response to salt stress (*Che-Othman et al., 2017*; *Jacoby et al., 2011*; *Yang and Guo, 2018*). Additionally, aligned with our results, multiple studies have reported a positive correlation between salt tolerance and levels of secondary metabolites and amino acids with osmoprotectant properties associated with lowering osmotic stress (*Krishnamurthy and Bhagwat, 1990*; *Petrusa and Winicov, 1997*; *Tari et al., 2010*). Our results suggest that salt exposure induces decoherence of gene expression in some transcripts, that this decoherence results from a restructuring of the gene expression network, and this restructuring can allow salt stress tolerance, providing a potential molecular mechanism underlying this tolerance.

## Selection on organismal traits

In addition to gene expression data, we also collected phenotypic data for 13 organismal traits in normal and saline conditions, which provides us with the opportunity to examine how salt stress

influences complex phenotypes and identify connections between variation in traits with fitness consequences and underlying patterns of gene expression (*Figure 4*). Since these traits were measured on different scales, we estimated variance-standardized selection gradients on these traits, focusing only on seven traits that were not strongly correlated to limit the contribution of indirect selection (Pearson correlation coefficient <0.6; *Figure 4—figure supplement 1*). We identified three traits – leaf osmotic potential (LOP), chlorophyll a content (Chl_a), and flowering time (FT) – that displayed different selection patterns in normal versus saline environments (*Figure 4*; *Supplementary file 9*).

Leaf osmotic potential is a key trait associated with water transport in plants, and which decreases with increasing salinity, leading to low water uptake by the plant. This trait was under positive directional selection in the control environment but under stabilizing selection under saline conditions, which suggests that an optimal LOP is important under salt stress (*Figure 4*). Furthermore, we found Chl_a content at the reproductive stage to be under negative directional selection in saline conditions. Although ion toxicity has previously been shown to reduce chlorophyll content (*Ashraf and Bhatti, 2000*; *Taïbi et al., 2016*), our results showed that this reduction is associated with increased survival and reproductive fitness, consistent with the general trend for reduced photosynthesis under salinity stress. We also found that FT was under positive directional selection (selection for later flowering) in normal conditions but under stabilizing selection in saline conditions (*Figure 4*). Moreover, FT was significantly reduced under saline conditions (two-tailed paired t-test p=0.0012; *Figure 4—figure supplement 2*), implying that salt stress selects for an earlier flowering compared to that in the normal wet paddy.

## Genetic architecture of gene expression variation

To dissect the genetic architecture of gene expression variation, we identified expression quantitative trait loci (eQTLs) and examined whether and how selection acts on these eQTLs (*Figure 5*). We identified *cis*- and *trans*-eQTLs (FDR <0.001) regulating expression of 3065 and 3277 genes in normal and salinity stress conditions, respectively (*Supplementary file 10*). The median number of *cis*- and *trans*-eQTLs regulating the expression of a gene were similar between environments (median eQTL per gene *cis*-normal=14, *cis*-saline=13, *trans*-normal=5, *trans*-saline=4). We observed that 49.64% (29,623 of a total 59,669) of *cis*-eQTLs were common in both environments, as compared to 18.62% (25,528 of a total 137,100) of *trans*-eQTLs; this result was robust to FDR cutoff (FDR of 0.01 and 0.05). This is consistent with previous observations from studies on other species that have found 48–77% overlapping *cis*-eQTLs and 9–60% common *trans*-eQTLs across environments (*Smith and Kruglyak, 2008*; *Snoek et al., 2012*; *Sterken et al., 2023*). Moreover, comparing the effect sizes of the two categories of eQTLs, we found that *trans*-eQTLs explain more variation in transcript abundance as compared to *cis*-eQTLs in both environments (two-tailed t-test $p<10^{-16}$; mean effect size: *cis*-normal=0.77, *trans*-normal=1.04, *cis*-saline=0.80, *trans*-saline=1.08). This further indicates that *trans*-eQTLs might be more environment-specific than *cis*-eQTLs. We tested this explicitly by identifying loci showing gene-environment interaction (G×eQTL) and found that at FDR of 0.05, *cis*-eQTLs constituted merely 0.28% (142 of 50718) of the total identified G×eQTLs.

We identified eQTL hotspots, which are regions of the genome that are associated with expression variation of a large number of genes (*Qu et al., 2018*). These regions can occur either due to low amounts of recombination (high linkage disequilibrium- LD) or because they contain master-regulators that pleiotropically control expression of multiple functionally-associated genes (*Hammond et al., 2011*; *Kliebenstein, 2009*; *West et al., 2007*). To account for LD and identify regions likely to contain master-regulators, we chose to focus on the subset of genes regulated by the lead-SNPs (SNP with the most significant association) in a given 100 kb region. Through this approach, we identified 11 *trans*-eQTL hotspots (number of unique genes >30) in saline conditions, but none in normal conditions (*Figure 5—figure supplement 1*; *Supplementary file 11*). These results indicate that natural variation in gene regulation under stress conditions may be dependent on a few master-regulators, as has been shown for drought stress in rice and maize (*Kuroha et al., 2017*; *Liu et al., 2020*). Interestingly, one of the hotspots, Chr7: 25.9–26.0 Mb, influenced the expression of a disproportionately high number of genes (191 genes) and contains the gene *OsFLP*, a R2R3 myb-like transcription factor that regulates stomatal development (*Wu et al., 2019*). *OsFLP* has recently been associated with salt tolerance in rice (*Qu et al., 2022*).

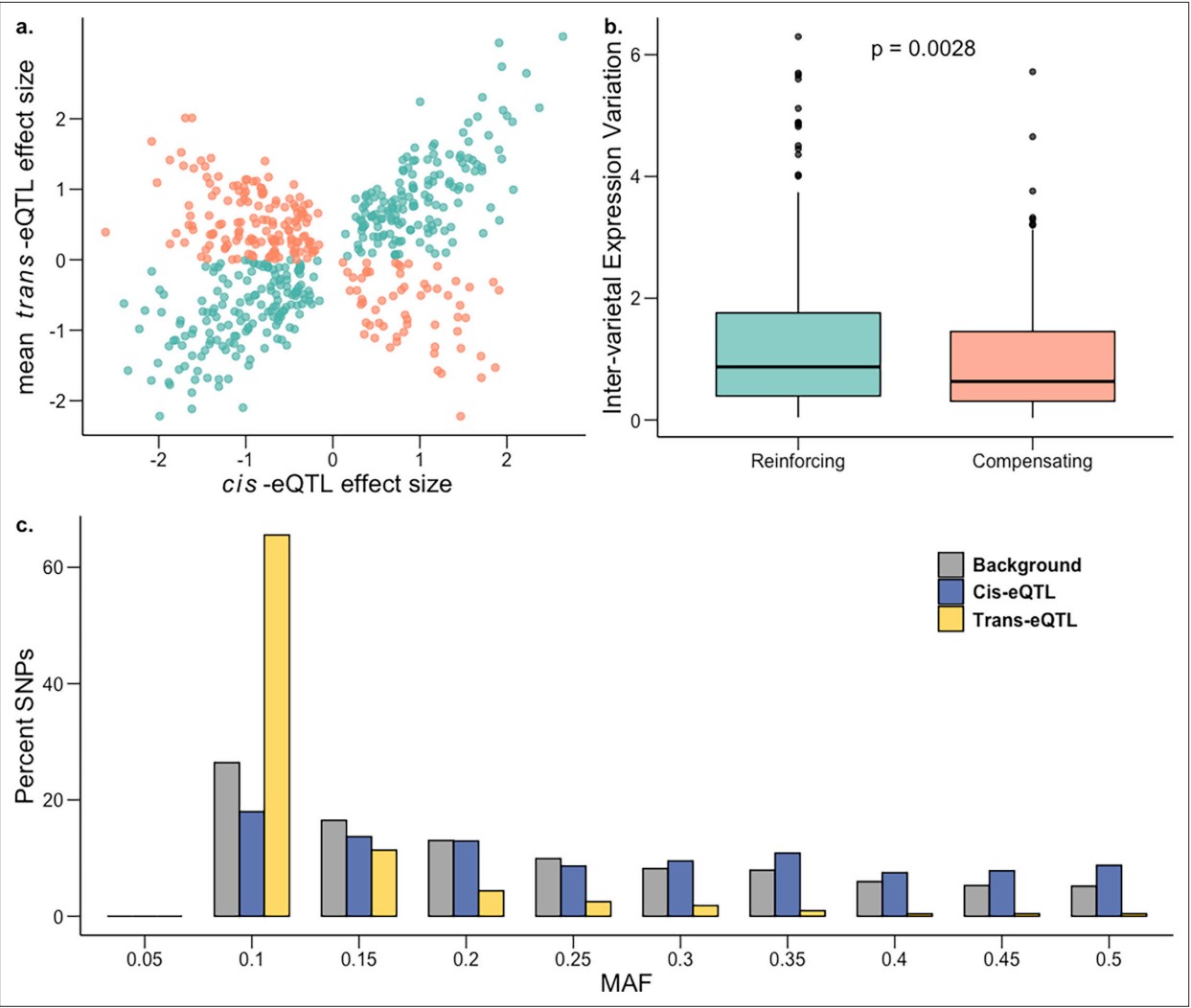

**Figure 5.** Genetic architecture of gene expression variation during salt stress. (**a**) Effect sizes of genes with both *cis* and *trans* factors under saline conditions showing excess of reinforcing cis-trans (teal) in comparison to compensating cis-trans (salmon). (**b**) Inter-varietal variation in gene expression for genes under compensating control is significantly lower than for those under reinforcing control; one-sided Mann-Whitney p=0.0028. c, Frequency distribution of MAF (minor allele frequency) for *cis*-eQTLs (blue) and *trans*-eQTLs (yellow) in saline conditions against the genome-wide background (gray).

The online version of this article includes the following figure supplement(s) for figure 5:

**Figure supplement 1.** *Trans*-eQTL hotspots in normal (**a**) and salinity stress (**b**) conditions.

**Figure supplement 2.** Compensating and reinforcing *cis-trans* effects in normal conditions.

**Figure supplement 3.** Frequency distribution of MAF (Minor Allele Frequency) for *cis*-eQTLs (blue) and *trans*-eQTLs (yellow) in normal conditions against the genome-wide background (gray).

It has been shown previously that when a gene is regulated via both *cis* and *trans* factors, their effects tend to drive target gene expression in opposite directions, canceling the combined effect on expression. This is commonly referred to as *cis-trans* compensation, which is thought to arise due to stabilizing selection that maintains similar gene expression over evolutionary timescales in the face of new mutations (*Coolon et al., 2014*; *Goncalves et al., 2012*; *Landry et al., 2005*; *Lovell et al., 2018*). Contrary to this expectation, we found that more than half genes appear to have a reinforcing (*cis* and *trans* effect in the same direction) rather than a compensatory pattern (two-tailed proportion z-test p=1.65 x 10$^{-11}$; *Figure 5a*; *Supplementary file 12*). Among the 524 *cis-trans* co-occurring genes in saline conditions, 317 were reinforcing (~60.5%) and 207 were compensatory (39.5%). This pattern held for normal conditions as well (*Figure 5—figure supplement 2a*; *Supplementary file 12*). Since

these reinforcing genes have *cis* and *trans* variants acting in the same direction, it indicates that the expression of these genes are expected to either increase or decrease over evolutionary timescales, indicating the presence of directional selection.

We hypothesized that this excess of reinforcing rather than compensatory effects could be driven by the extensive diversity among rice landraces that arose from crop diversification following domestication (*Meyer and Purugganan, 2013*). To test this hypothesis, we examined whether genes under *cis-trans* reinforcement showed evidence of higher inter-varietal variation in gene expression (population-wide variance between mean accession expression levels) as compared to genes under *cis-trans* compensation. Supporting our hypothesis, we found significantly higher inter-varietal expression variation for genes under *cis-trans* reinforcement in saline (*Figure 5b*; Mann-Whitney *U*-test=0.002; mean-inter-varietal expression variation compensating = 1.12, reinforcing = 1.27) and in normal (*Figure 5—figure supplement 2b*) conditions. Furthermore, it has been suggested that *cis-trans* compensation/reinforcement can arise due to the genetic fixation of compensating/reinforcing *trans*-regulatory variants, potentially due to epistasis (*McManus et al., 2014*; *Signor and Nuzhdin, 2018*), which could lead to elevated linkage disequilibrium (LD) between the *cis*- and *trans*-regulatory variants acting on a gene. This was indeed the case: estimated LD between all pairs of *cis*- and *trans*-regulatory variants for the identified compensating and reinforcing genes was significantly higher as compared to the background LD (mean $r^2$ compensating: 0.33, reinforcing: 0.36, background = 0.056; two-tailed permutation test p=0.0019).

To examine the pattern of past selection acting on *cis*- and *trans*-eQTLs (before, during and after rice domestication, including the crop diversification phase) we plotted the folded site-frequency spectrum (SFS) of the minor allele frequencies and inferred the relative strength of historic selection (*Joly-Lopez et al., 2020*) We found that *trans*-eQTLs for both environments had been under strong purifying selection, with the SFS being significantly left-shifted compared to background SNPs (two-tailed t-test $p<10^{-16}$; mean MAF background = 0.20, *trans* normal = 0.0936, *trans* saline = 0.0910). Not surprisingly, given their highly pleiotropic nature, this effect was more pronounced for *trans*-eQTLs regulating multiple genes (two-tailed t-test $p<10^{-16}$; normal mean MAF in unique *trans*-eQTLs=0.099, multiple *trans*-eQTL=0.073; saline mean MAF in unique *trans*-eQTLs=0.097, multiple *trans*-eQTL=0.071). In contrast to *trans*-eQTLs, we found that *cis*-eQTLs in both normal and saline conditions had potentially been under balancing selection (*Figure 5C*, *Figure 5—figure supplement 3*) with the SFS being significantly right-shifted for *cis*-eQTLs relative to background SNPs (two-tailed t-test $p<10^{-16}$; mean MAF background = 0.20, *cis* normal = 0.238, *cis* saline = 0.237). Moreover, for both conditions, the estimated nucleotide diversity (π) for 100 kb regions flanking *cis*-eQTLs was also significantly higher as compared to genome-wide 100 kb blocks (two-tailed t-test $p<10^{-11}$; mean π background = 0.286, *cis* normal = 0.297, *cis* saline = 0.297).

Comparing patterns of past selection for *cis*-eQTLs specific to each condition revealed no significant difference between the normal and saline environments. However, we did observe *trans*-eQTLs in saline conditions to be under stronger purifying selection (two-tailed t-test p=0.017). Taken together, this indicates that *cis*- and *trans*-eQTLs are under different selection regimes in rice, and that purifying selection is stronger on *trans*-eQTLs regulating gene expression under salt stress conditions, giving us insights into the macroevolutionary dynamics of gene-expression variation.

## Discussion

Gene expression is a key link in the chain of organismal responses to environmental challenges. In this study, we used a systems genomics approach to examine genome-wide transcript levels in rice in both normal and moderately saline field conditions to examine the evolutionary response and genetic architecture of gene expression under salinity stress. We found selection on expression of a set of genes, and for some genes, selection on expression differed among environments, indicating that salinity can select for changes in gene expression. We saw that the genetic regulatory pathways can be modified by salinity, as indicated by decoherence, which provides a potential genetic mechanism underlying salinity tolerance. We found limited evidence for trade-offs given a lack of antagonistic pleiotropy in the fitness effects of expression compared across environments, and we also found that there did not appear to be strong cis-trans compensation in gene regulation. These results provide us with insights into the fitness consequences and genetic architecture of gene expression under salt stress in rice, as we discuss below.

One of the primary goals of this study was to characterize selection on gene expression and how this selection varies between saline and normal conditions. Before discussing our results, it is important to note that our conclusions are influenced by the way in which selection differentials are standardized. Selection differentials are estimated through regression coefficients in the relationship between trait values and fitness (*Lande, 1979*). Because traits are often measured on different scales, selection differentials are usually standardized to allow for meaningful comparisons and interpretation. The most common method of standardization is variance standardization, in which traits are standardized to a mean of zero and a standard deviation of one (*Lande and Arnold, 1983*). With this approach, the selection differential, multiplied by the heritability, gives the expected response to selection in standard deviation units. So while this is a valid approach, there is concern that when variance-standardized selection differentials are interpreted as reflecting the 'strength' of selection based on their magnitude, they can be misleading because trait variance can vary among traits and environments, such that the trait variance and the magnitude of selection are conflated (*Hereford et al., 2004*). Instead, the use of mean-standardized selection estimates have been suggested (*Hereford et al., 2004*; *Matsumura et al., 2012*). However, it has not been widely recognized that mean-standardized selection estimates can also be biased if mean trait values vary consistently across conditions, such that magnitude of selection is conflated with trait means. We show here that this was indeed the case: the mean gene expression values differed consistently between saline and normal conditions, leading to biases when comparing the strength of selection across treatments when mean-standardized selection differentials were used. Specifically, using mean-standardized selection coefficients, we found selection on gene expression to be stronger under normal conditions, whereas unstandardized and variance-standardized selection coefficients indicated stronger selection under saline conditions. This result indicates that it is important for researchers to carefully consider the advantages and potential drawbacks of both variance-standardized and mean-standardized, as well as unstandardized, selection differentials in making decisions about which to use in any particular system and depending of the goals of the study, but also to consider these ideas in interpreting selection differentials as reflecting the strength of selection.

Because we found that both variance-standardized and mean-standardized selection gradients were biased, as explained above, we draw our conclusions about patterns of selection using unstandardized selection gradients. We believe this approach to be appropriate given that gene expression values are measured on the same scale and normalized for every transcript, and our goal is to identify transcripts under selection and compare selection across treatments. Using unstandardized selection coefficients, our study shows that most genes appear to have nearly neutral levels ($|S|<0.1$) of selection on transcript levels in both normal and saline environments. This is consistent with previous studies conducted in various plant and non-plant species, which have shown that most traits (including transcript abundance) are under very weak selection, and only a few traits experience strong selection at microevolutionary timescales (*Ahmad et al., 2021*; *Hoekstra et al., 2001*; *Kingsolver et al., 2001*). There does seem to be a tendency, however, towards stronger positive over negative directional selection on gene expression, meaning that higher expression of a majority of genes is generally associated with higher fitness. Looking at quadratic selection, we found evidence for both stabilizing and disruptive selection on gene expression. Our results are consistent with the expectation of stabilizing selection being more common than disruptive selection in normal conditions (*Kingsolver et al., 2001*), and reinforce the idea that disruptive selection is more widespread under stress conditions (*Ahmad et al., 2021*), potentially reflecting the prevalence of frequency- and density-dependent competition for resources under stress (*Kingsolver and Pfennig, 2007*).

Comparing the distribution of selection coefficients, we further found that selection was stronger in moderate stress saline field conditions compared to normal wet paddy field conditions, indicating that salinity stress induced increased selective pressure on gene expression at microevolutionary timescales. Although exposure to salinity stress increased the strength of selection on gene expression, this increase was relatively low compared to what has been reported for drought stress in rice (*Agrawal and Whitlock, 2010*; *Groen et al., 2020*; *Kondrashov and Houle, 1994*). This could potentially be attributed to the levels of stress experienced by the plants – the drought stress treatment was more severe in terms of fitness after stress exposure (*Agrawal and Whitlock, 2010*; *Groen et al., 2020*; *Kondrashov and Houle, 1994*) in comparison to the moderate salinity stress in this study – but further work is needed to examine whether there is indeed a relationship between stress intensity and

selection strength. While this work provides insight into the direction of selection and the predicted degree of evolutionary change under the specific conditions of this study, more work would need to be done over multiple generations and under additional conditions to make more realistic predictions of microevolutionary change in the field.

Our work contributes to an ongoing debate regarding whether the intensity of selection on gene expression increases under stress. We found in this study of rice and salinity, and in a prior study on drought (*Groen et al., 2020*), that directional selection was greater under stress compared to normal conditions. This finding is consistent with prior studies on other systems like fruit flies (*Drosophila melanogaster*; *Groen et al., 2020*; *Jasnos et al., 2008*; *Kondrashov and Houle, 1994*) and yeast (*Groen et al., 2020*; *Jasnos et al., 2008*; *Kondrashov and Houle, 1994*), but contrasts with a recent study in *D. melanogaster* reporting no change in selection due to increasing levels of nutritional stress (*Arbuthnott and Whitlock, 2018*). Although we found selection to be stronger under saline conditions using total filled grain number as a measure of fecundity, a caveat to selection estimates is that they are sensitive to the proxy of fitness. Our results also support prior findings in rice (*Calic et al., 2022*; *Groen et al., 2020*) and in *Boechera stricta* (*Anderson et al., 2013*) that conditional neutrality may be much more common than antagonistic pleiotropy, although the lack of antagonistic pleiotropy could be due to lack of trade-offs in the years evaluated or to low power to detect small-effects trade-offs. Further, both conditional neutrality and antagonistic pleiotropy have been shown to underlie local adaptation (*Anderson et al., 2013*; *Wadgymar et al., 2017*), following which our findings indicate that it may be feasible to breed salinity tolerant rice varieties without a yield penalty under non-saline conditions, though further work is needed to verify this.

We examined the genetic architecture of gene expression variation as well, and found that *trans*-eQTLs rather than *cis*-eQTLs are primarily associated with rice gene expression under salinity stress, potentially via a few master-regulators. *Trans*-eQTLs may be important in environment-dependent gene expression changes, while *cis*-eQTLs may be more robust to environmental changes as has been observed in other species (*Smith and Kruglyak, 2008*; *Snoek et al., 2012*; *Sterken et al., 2023*). This can be attributed to *trans*-eQTLs' larger mutational target along with a larger effect of drift than of positive selection in fixing them. These results further corroborates with other studies that have found a more dominant effect of *trans*-eQTLs on within-species variation than between-species variation (*Emerson et al., 2010*; *Metzger et al., 2016*; *Wittkopp et al., 2008*).

We found that *cis*- and *trans*-eQTLs show different patterns of selection, with *cis*-eQTLs showing evidence for balancing selection and *trans*-eQTLs showing purifying selection. We see this pattern for rice, a largely selfing species, in contrast to outcrossing species where both *cis*- and *trans*-eQTLs have been found to be under purifying selection (*Hernandez et al., 2019*; *Josephs et al., 2015*). Future studies will be needed to investigate whether the mating system has an influence on the pattern of natural selection acting on variants regulating gene expression. Finally, and contrary to expectations, we show that under saline conditions, *cis-trans* reinforcement is more prevalent than *cis-trans* compensation. This result may be driven by rice domestication and subsequent population diversification. Additionally, we find significantly elevated levels of LD among the *cis*- and *trans*-regulatory variants for compensating and reinforcing genes, indicating the role of genetic fixation as an underlying mechanism for *cis-trans* compensation/reinforcement.

Systems genomic approaches provide insights into large-scale patterns across genome-wide data over multiple scales. This approach can also identify possible genes or genetic pathways that may prove critical in various processes. Our analysis, for example, shows that many of the potential antagonistically pleiotropic genes are involved in photosynthesis and metabolic processes. Importantly, we found that the circadian rhythm pathway was under positive directional selection (selection for increased expression) in saline conditions and that the genes involved in circadian rhythm showed significant decoherence between the two environments. These findings make sense in light of the fact that circadian rhythm has been shown to regulate salt tolerance and flowering time in rice (*Liang et al., 2021*; *Wei et al., 2021*), so responses to salt stress may alter the expression patterns of genes in the circadian rhythm network, leading to decoherence. In further support of this idea of a link between salt tolerance and circadian rhythm, our selection analyses on physiological traits shows that salt stress leads to selection for earlier flowering. Additionally, our decoherence analyses identified transcripts in important pathways related to plant responses and potentially tolerance to excess salts, including the carbon metabolic pathways (glycolysis and tricarboxylic acid cycle), and

accumulation of secondary metabolites and sugar moieties with osmoprotectant properties (*Jacoby et al., 2011*; *Krishnamurthy and Bhagwat, 1990*; *Petrusa and Winicov, 1997*; *Tari et al., 2010*; *Yang and Guo, 2018*). But as is with such analyses, detection of population-level correlations are temporally biased and may fail to detect the transient environmentally responsive links resulting in false negatives (*Cai and Des Marais, 2023*) and leading to the failure to detect temporally key processes.

Other genes that are beneficial only in the saline environment include a cyclophilin-encoding transcript (*OsCYP2*), which has been shown to confer salt tolerance in rice (*Ruan et al., 2011*). Moreover, we also identified eQTL variants regulating expression of two photosynthesis related antagonistically pleiotropic transcripts (*PSAN* and *CRR7*) in the normal environment. Finally, our *trans*-eQTL analysis identified a hotspot on chromosome 7 that contains *OsFLP*, a R2R3 myb-like transcription factor that regulates stomatal development (*Wu et al., 2019*), and appears to be involved in salt tolerance in rice (*Qu et al., 2022*). Together, these loci are possible new targets for functional and translational studies of salinity stress in rice. Coupled with the insights gained by a systems genomic approach in inferring large scale patterns from genome-wide information, this integration of data across multiple scales across a rice population has allowed us to provide an integrated examination of the molecular and genetic landscape underlying adaptive plant salinity stress responses.

# Materials and methods
## Plant material

Domesticated rice is primarily classified into two distinct genetic subgroups, *O. sativa* ssp. *indica* and *O. sativa* ssp. *japonica*. These subgroups are grown in sympatry and are often recognized as subspecies, given the reproductive barriers between them (*Nadir et al., 2018*). Further analyses have identified a widely accepted classification consisting of genetically distinct varietal groups namely, *indica*, *aus/circum-aus*, *aromatic/circum-basmati*, *tropical japonica*, and *temperate japonica* (*Garris et al., 2005*). For this study, a total of 130 *O. sativa* ssp. *indica* (including *indica* and *circum-aus* groups) and 65 *O. sativa* ssp. *japonica* (including *circum-basmati*, *tropical japonica*, and *temperate japonica*) accessions were selected, including traditional varieties/landraces and three additionally replicated salt-sensitive and -tolerant test varieties. We focused our analyses on *O. sativa* ssp. *indica* since it is the predominant global varietal group (*Supplementary file 13*). Seeds were obtained from the International Rice Genebank Collection (IRGC) at the International Rice Research Institute (IRRI) in the Philippines, and from a 2016 bulk seed collection obtained from plants grown under normal (wet and non-saline) conditions at IRRI during the course of a previous study (*Groen et al., 2020*).

## Field experiment

The field experiment was conducted in the dry season of 2017 at IRRI in Los Baños, Laguna, Philippines. Seeds from each accession were sown on December 16, 2016, and seedlings were then transplanted into the experimental fields at 17 days after sowing (DAS), on January 5, 2017. The field experiment was conducted across two locations: site L4 (14°09'34.6"N 121°15'42.4"E) was prepared as the non-salinized 'normal' environment and site L5 (14°09'35.2"N 121°15'42.5"E) as the salinized environment, following *Groen et al., 2020*. Within each field environment, there were three blocks and three replicates of each genotype (accession), with each genotype planted once per block in a random location. Each plant was planted in a single-row with 0.2-m × 0.2-m spacing between them for a total of one focal plant and seven neighboring plants (included in the experiment) per plot. Each experimental plot included the accessions NSIC Rc 222 and NSIC Rc 182 that served as border rows (*Supplementary file 13*). The application of salt in site L5 started on January 19, 2017, when the plants were 31 days old. The salinity level was monitored by recording electrical conductivity (EC), using EC meters installed in each of the parcels at a depth of 30 cm. The EC levels were recorded twice per day until reaching an EC = 6 dSm$^{-1}$ and then recorded daily. The salinity levels were then maintained at 6 dSm$^{-1}$ (considered mild to moderate salinity stress) until maturity. Management and maintenance of the fields included the application of basal fertilizer, spraying of insecticides against thrips and removal of plants potentially infected with rice tungro virus.

## Tissue collection for transcriptome sequencing

Leaf sampling was performed as previously described (*Groen et al., 2020*). Briefly, leaf collection in the non-saline and saline field was done from 10:00 hr to 12:00 hr at 38 DAS (8 days after the beginning of the salt treatment) in the non-saline and saline field from 10:00 hr to 12:00 hr. Both fields were sampled simultaneously and with individuals within a block collected in the same order. For each sample, about 10 cm of leaf length were cut into small pieces and placed in chilled 5 mL tubes containing 4 mL of RNALater (Thermo Fisher Scientific) solution for RNA stabilization and storage. Leaf samples from each of the 5 ml tubes were then transferred into pairs of 2 mL tubes (one for processing and one for backup), then stored at −80 °C.

## Yield harvesting and panicle trait phenotyping

A total of 780 plants were harvested individually and labeled such that the yield of all plants used for each type of measurement (mRNA sequencing, phenotypic measurements) was known. Individual seeds collected were further categorized as filled, partially filled, and unfilled, using manual assessment and a seed counter (Hoffman Manufacturing). Panicle length measurements and panicle trait phenotyping (PTRAP) of 30 seeds were also performed.

## Functional trait phenotyping

In addition to yield-related measurements, we collected data on a number of physiological, morphological, and phenological traits to assess differences between rice accessions in response to soil salinity. In both the non-saline and saline fields, we recorded leaf osmotic potential (LOP) in the vegetative stage, performed chlorophyll analysis based on 1 mg leaf samples and measured ion content (analysis of sodium, $Na^+$, and potassium, $K^+$, analysis) based on 20 mg of leaf samples in both the vegetative and reproductive stages. We also measured plant height for growth rate (measured once a week until maturity). Flowering time was recorded as the day on which 50% of plants in a plot flowered; these plants included the focal plant and its seven neighboring plants. Whole plants were both harvested at the vegetative stage and at maturity to measure wet and dry biomass.

## Extraction of total RNA for library construction

Leaf samples stored at −80 °C were thawed at room temperature briefly and excess RNALater was removed. Tissue samples were then flash-frozen in liquid nitrogen and ground using a TissueLyser II (QIAGEN). After this, total RNA was extracted using the RNeasy Plant Mini Kit according to manufacturer's protocol (QIAGEN) and eluted in nuclease-free water. The integrity of total RNA was assessed by agarose gel electrophoresis, and RNA from a random subset of samples was further assessed by Agilent TapeStation (Agilent Technologies). RNA concentration was quantified on a Qubit (Invitrogen). Samples were stored at −80 °C until library preparation.

## RNA-seq library preparation and sequencing

Library preparation for 780 samples was performed as described in *Groen et al., 2020* and followed a plate-based 3′-end mRNA sequencing (3′ mRNA-seq) protocol. Briefly, total RNA from each sample was transferred individually into 96-well plates and normalized to a concentration of 10 ng in 50 µL nuclease-free water. Then, mRNA samples were reverse-transcribed using Superscript II Reverse Transcriptase (Thermo Fisher Scientific) and cDNAs were amplified using the Smart-seq2 protocol (*Picelli et al., 2013*) with modifications (*Groen et al., 2020*). This resulted in multiplexed pools of 96 samples, where 48 samples were from the non-saline field environment and 48 samples were from the same plot number in the saline field environment. Each pool was used for library preparation with the Nextera XT DNA sample prep kit (Illumina), returning 3′-biased cDNA fragments, similar to the Drop-seq protocol (*Macosko et al., 2015*). The resulting cDNA libraries were then quantified on an Agilent BioAnalyzer and sequenced at the NYU Genomics Core on an Illumina NextSeq 500 with the configuration HighOutput 1x75 base pairs (bp) and the settings: Read 1 of 20 bp (bases 1–12, well barcode; bases 13–20, unique molecular identifier [UMI]) and Read 2 of 50 bp. Raw sequence reads have been submitted to the SRA (BioProject PRJNA1010833).

## RNA-seq data processing and data normalization

3'mRNA-seq read data were processed as previously described in *Groen et al., 2020*. Briefly, Drop-seq tools v1.12 (RRID:SCR_018142; https://github.com/broadinstitute/Drop-seq) and Picard tools v2.9.0 (https://broadinstitute.github.io/picard/) were used to generate the metadata. The reference genome, Nipponbare IRGSP 1.0 (ftp://ftp.ncbi.nlm.nih.gov/genomes/all/GCF_001/433/935/GCF_001433935.1_IRGSP-1.0), and annotations were indexed with STAR v020201 (*Dobin et al., 2013*). Prior to generating read counts, raw reads were converted from FASTQ to unaligned BAM format using Picard tools FastqToSam before being processed using the unified script for a FASTQ starting format. After this, digital gene-expression matrices displaying either UMI or raw read counts with transcripts as rows and samples as columns containing counts from reads with 96 expected sample barcodes were produced using the DigitalExpression utility. Sample barcodes corresponding to beads never exposed to rice total RNA were filtered out based on low numbers of transcribed elements as described previously (*Groen et al., 2020*; *Macosko et al., 2015*). Rice individuals that ended up being discarded due to low numbers of transcribed elements, were sequenced again using another library.

UMI counts per sample were normalized through dividing by the total number of detected UMIs in that sample and multiplying by $1\times10^6$ to obtain transcripts per million. The resulting data matrices were then merged into one digital gene-expression super-matrix, containing transcripts-per-million expression data for all samples. Elements with very low transcription levels (transcript models with a sigma signal <20) were discarded, after which a robust normalization was conducted using an invariant set normalization protocol within the DChip utility v2010.01 (*Li and Wong, 2001*). All downstream analyses were done in log-space, using normalized expression levels ($\log_2$[normalized transcripts-per-million value +1]) of transcribed elements estimated using R v3.4.3 (*R Development Core Team, 2013*; *Robinson et al., 2010*). In a final step of filtering, transcripts that were not detected in at least 10% of individuals across our populations and did not derive from protein-coding genes on nuclear chromosomes were removed prior to performing subsequent analyses.

## Quantitative genetics of fecundity and gene expression

All downstream analyses were done using R v3.4.3 (*R Development Core Team, 2013*; *Robinson et al., 2010*). The effect of genotype (G), environment (E) and genotype-by-environment (G×E) on fecundity (number of filled grains) was assessed using two-way ANOVA with E as a fixed effect, and G and G×E as random effects (https://www.angelfire.com/wv/bwhomedir/notes/anova2.pdf). Essentially, the between sum of squares for G, E, and G×E was estimated using a one-way ANOVA with the aov function, which was then used to estimate the F-statistic. The significance of each term was determined using the F-tests with pf function. Fecundity, averaged by genotype, was further compared between the environments using a two-tailed paired t-test. Variation in gene expression was partitioned similarly using the same model as above, and significance of each term was tested using F-tests via a mixed-model ANOVA (*Howell, 1997*). Multiple testing was controlled using a False-Discovery Rate (FDR) of 0.001. Broad-sense heritability was estimated as $H^2 = \sigma^2_G /(\sigma^2_G + \sigma^2_{GE}/e + \sigma^2_E/re)$, with $\sigma^2_G$, $\sigma^2_E$ and $\sigma^2_{GE}$ as the variance explained due to G, E, and G×E; e and r represent the number of environments and number of replicates per environment, respectively. Inter-varietal differences in gene expression was estimated for each environmental condition as the population-wide variance between accession mean expression levels.

## Univariate and multivariate selection analyses

Univariate selection differentials consider each trait separately and represent the total strength of selection acting on a trait (*Lande, 1979*; *Lande and Arnold, 1983*). Fitness was estimated as fecundity, which was the total number of filled rice grains per individual. Unstandardized linear selection differentials (*S*) on gene expression were estimated as coefficients of linear regression with fecundity as the dependent variable in each regression and transcript abundance and block as the independent variables. Unstandardized quadratic selection differentials (*C*) were estimated similarly as twice the coefficients of quadratic regression for transcript abundance and fecundity (*Conner and Hartl, 2004*; *Stinchcombe et al., 2008*). Regressions were performed using the lmer function (*Bates et al., 2015*). Using these we then estimated the variance-standardized, and mean-standardized selection differentials (*Hereford et al., 2004*; *Lande and Arnold, 1983*). Data preparation included filtering out individuals with zero fecundity followed by normalizing fecundity fitness by mean fitness. To satisfy

normality and remove noise inherent in expression data, transcripts with expression values more than 3 standard deviations from the mean were removed, which affected fewer than 1% of individuals. Selection differentials were estimated for transcripts that were expressed in at least 20 individuals and were estimated separately in each of the two environments. Multiple testing was controlled using Bonferroni correction (*Bland and Altman, 1995*) using the p.adjust function.

Multivariate selection gradients represent the strength of direct selection acting on each trait, after removing indirect selection caused by correlations with other traits (*Lande and Arnold, 1983*). Principal component analysis (PCA) was performed on transcript abundance using the prcomp function in R (*R Development Core Team, 2013*; *Kassambara, 2017*) and PCs explaining over 0.5% of variance in each environment were chosen for multivariate selection analyses. Linear (β) and quadratic (γ) selection gradients were estimated as coefficients of multiple regression with the normalized fecundity fitness as the dependent variable and the PCs as the independent variables (*Conner and Hartl, 2004*). We then chose the top one percent of transcripts showing the highest loadings of the PCs (n=182) and counted the number of transcripts showing evidence for positive directional selection (same directionality between loading of the transcript on the PC and univariate selection differential (*S*) acting on the transcript estimated above) versus negative directional selection (opposite directionality between loading of the transcript on the PC and univariate selection differential (*S*) acting on the transcript estimated above) and based on the majority assigned a directionality to the selection gradients on PCs. Variance-standardized multivariate selection gradients were estimated for functional traits without strong correlation to avoid collinearity among traits (*Lande and Arnold, 1983*; *Presotto et al., 2019*), using a Pearson correlation coefficient <0.6 as threshold, which was estimated using the cor function in R (*R Development Core Team, 2013*).

## Selection analyses on gene ontology biological processes

Gene Ontology (GO) term annotations for rice genes/transcripts were downloaded from Monocots PLAZA 5.0 (*Van Bel et al., 2022*). All fourth-level biological-process terms were downloaded using GO.db v3.15.0 (*Carlson, 2022*) and only these terms were considered for further analyses. Next, terms with fewer than 20 transcripts in our dataset were filtered out to minimize redundancy, leaving a total of 670 terms and 10,235 associated transcripts. The selection strength on a biological-process term was estimated as the median selection strength of all transcripts annotated with that term. A term was considered to be under significantly stronger selection compared to the transcriptome-wide median if the median strength of selection for a term was over the transcriptome-wide median selection strength by at least the 95% confidence interval for the selection strength of that term. GO enrichments were done using ShinyGO (*Ge et al., 2020*).

## Regulatory decoherence analyses

To examine regulatory decoherence in rice, a recently developed method, CILP (Correlation by Individual Level Product), was used (*Lea et al., 2019*). CILP first estimates the correlation of a phenotype (transcript expression in our case) within each individual in the sample and then using linear model or linear mixed effect model tests for associations between this estimate and a fixed effect predictor variable (environment in our case), while controlling for covariates. Since CILP calculates product correlations for all possible pairs of genes, only transcripts with a selection strength greater than 0.1 (|*S*|>0.1) in at least one environment with expression greater than 0 in at least 50% individuals were included to reduce the dimensionality (leaving 2318 transcripts). Multiple testing was controlled using a False-Discovery Rate (FDR) of 0.05.

## Genotype data and SNP calling

Raw FASTQ files were downloaded from the Sequence Read Archive (SRA) website under BioProject PRJEB6180 for 14 accessions (*Wang et al., 2018*) and under Bioprojects PRJNA422249 and PRJNA557122 for 92 accessions (*Gutaker et al., 2020*; *Supplementary file 13*). Further, genomes of 19 accessions were re-sequenced, and submitted to the SRA (BioProject PRJNA1012700), leading to a total of 125 accessions for which genomic data was available (*Supplementary file 13*).

Raw reads were processed for quality control and adapter trimming using the bbduk program of BBTools version 37.66 (https://jgi.doe.gov/data-and-tools/bbtools/) using the options: minlen = 25 qtrim = rl trimq = 10 ktrim = r k=25 mink=11 hdist = 1 tpe tbo. The output from this program was

mapped to the reference genome *O. sativa* Nipponbare IRGSP 1.0 genome that was downloaded from NCBI Genome (https://www.ncbi.nlm.nih.gov/genome/?term=txid4530[orgn]) using bwa-mem2 v2.1 (*Vasimuddin et al., 2019*). PCR duplicates were marked and removed using the Picard tools version 2.9.0. SNPs were called using GATK HaplotypeCaller v4.2.0.0 to obtain a multi-accession joint SNP file. Only SNPs that were above 5 bp distance from an indel variant were taken. Next, SNPs were filtered using the recommended GATK hard filtering (*Van der Auwera et al., 2013*). Further, using vcftools v0.1.16 (*Danecek et al., 2011*), SNPs with at least 80% genotype calls and a minor allele frequency of 0.05 were retained (--max-missing 0.8 --maf 0.05). Since rice is a inbred species, we also removed any SNPs that displayed heterozygosity of over 5% identified using vcftools v0.1.16 – hardy (*Danecek et al., 2011*). Next, missing genotype calls were imputed and phased using Beagle v4.1 (*Browning and Browning, 2016*), and using vcftools v0.1.16 -m2 -M2 (*Danecek et al., 2011*) only biallelic SNPs were retained for further analyses. Finally, SNPs were randomly pruned such that one SNP per 1000 bp was retained using vcftools v0.1.16 –thin (*Danecek et al., 2011*), leaving a SNP dataset of 246,714 markers.

## G-matrix estimation and prediction of short-term phenotypic evolution

A G-matrix (*G*) representing the additive genetic variance and covariance was estimated for the principal component axes (PCs) by taking the eigengene. This was done by deploying GREML v1.94 (*Yang et al., 2011*). Although the principal components are by definition uncorrelated at the level of the individual replicate plants, they start showing genetic covariances when loading values of replicates from each genotype are averaged. Next, using the multivariate breeder's equation ($\Delta z = G\beta$), we predicted the response to short-term phenotypic selection ($\Delta z$) on the PCs.

## Association mapping

Association mapping was performed between the SNP markers and gene expression values recorded in the normal and saline environments. For this, the linear model in Matrix eQTL was used (*Shabalin, 2012*). The normalized gene expression values were averaged over the replicates in each environment separately and these averages were subsequently used to test for associations. The first five principal components (PCs) of the kinship matrix were estimated using GAPIT v3 (*Wang and Zhang, 2021*) and added as covariates to control for population structure. Associations were considered significant at a false-discovery rate (FDR)<0.001, and when significant were included in downstream analyses. Due to long stretches of homozygosity attributed to the highly inbred nature of rice, *trans*-eQTLs were defined as being on a different chromosome or at least 1 Mb away from a gene under its influence on the same chromosome; *cis*-eQTLs were defined as <100 kb away from an associated gene.

To identify significant G×eQTLs, we ran Matrix eQTL on the difference of expression in the normal and saline conditions (normal – saline) at FDR 0.05 (*Smith and Kruglyak, 2008*).

For each gene regulated by a SNP in cis or *trans*, a lead SNP was identified as the SNP with the most significant association within a 100 kb region. Furthermore, *trans*-eQTL hotspots were identified through analyzing the number of unique genes regulated by lead SNPs in a given 100 kb region. To detect genes under *cis-trans* compensation or reinforcement, the effect sizes of all lead *trans*-eQTLs were averaged for each gene and compared to the lead *cis*-eQTL for the same gene. Next, for these genes we estimated the mean proportion of individuals with opposite and same direction *cis-trans* allelic configuration. Genes were defined as compensating and reinforcing if they had at least 60% of individuals with opposite and same *cis-trans* allelic configuration, respectively. To examine the LD structure, we estimated $r^2$ using plink v1.9 (*Purcell et al., 2007*) between (1) all pairs of *cis-* and *trans*-variants for the identified compensating and reinforcing genes and (2) 1000 datasets of randomly selected SNP pairs (with equal numbers of variant pairs and a similar distribution of distances between the *cis-* and *trans*-variants as in 1). Next, we compared $r^2$ between (1) and (2) using a two-tailed permutation test.

To examine the patterns of past selection on eQTLs, we used the minor allele frequency (MAF) of the 246,714 SNP markers and compared these using the t.test function in R (*R Development Core Team, 2013*). Furthermore, we estimated the site-wise nucleotide diversity (π) and averaged it over 50 kb flanking regions around each *cis*-eQTLs (100 kb region total). We compared this π to the background nucleotide diversity, estimated as π averaged over 100 kb blocks throughout the genome

minus the 100 kb *cis*-eQTL region above. The difference in mean of nucleotide diversity was tested using the t.test function in R (*R Development Core Team, 2013*).

## Acknowledgements

We thank the New York University Center for Genomics and Systems Biology GenCore Facility for sequencing support, and the New York University High Performance Computing for supplying computational resources. We would also like to thank William Mauck and Nicholas Rogers for technical support, as well as Adrian E Platts and Jae Young Choi, for their help with post-sequencing data processing. We are grateful to current members of the Purugganan laboratory (particularly J Flowers, A Kurbidaeva, O Alam) and Elena Hamann at Fordham University (currently Heinrich Heine University Düsseldorf) for insightful discussions. SNG is supported by a grant from the Gordon and Betty Moore Foundation/Life Sciences Research Foundation (award no. GBMF2550.06 to SNG), startup funds from the University of California Riverside, and a grant from the National Institute of General Medical Sciences of the National Institutes of Health (award no. R35GM151194 to SNG). This work was funded in part by grants from the US National Science Foundation Plant Genome Research Program (IOS 1546218 and 2204374) and the Zegar Family Foundation.

## Additional information

### Competing interests

Irina Calic: Employee of Inari Agriculture Nv. The other authors declare that no competing interests exist.

### Funding

| Funder | Grant reference number | Author |
|---|---|---|
| National Science Foundation | IOS 1546218 | Michael D Purugganan |
| National Science Foundation | IOS 2204374 | Michael D Purugganan |
| Zegar Family Foundation | | Michael D Purugganan |
| Gordon and Betty Moore Foundation | GBMF2550.06 | Simon Niels Groen |
| Life Sciences Research Foundation | GBMF2550.06 | Simon Niels Groen |
| National Institute of General Medical Sciences | R35GM151194 | Simon Niels Groen |
| Canada Research Chairs | CRC-2021-00126 | Zoé Joly-Lopez |
| Natural Sciences and Engineering Research Council of Canada | RGPIN-2021-03302 | Zoé Joly-Lopez |
| University of California, Riverside | | Simon Niels Groen |

The funders had no role in study design, data collection and interpretation, or the decision to submit the work for publication.

### Author contributions

Sonal Gupta, Formal analysis, Visualization, Writing – original draft, Writing – review and editing; Simon Niels Groen, Data curation, Formal analysis, Writing – review and editing; Maricris L Zaidem, Andres Godwin C Sajise, Mignon Natividad, Kenneth McNally, Georgina V Vergara, Rakesh K Singh, Data curation, Writing – review and editing; Irina Calic, Investigation, Writing – review and editing; Rahul Satija, Data curation; Steven J Franks, Supervision, Writing – review and editing; Zoé Joly-Lopez,

Data curation, Supervision, Writing – review and editing, Conceptualization, Investigation, Methodology; Michael D Purugganan, Conceptualization, Supervision, Funding acquisition, Writing – review and editing

**Author ORCIDs**
Sonal Gupta ⓘ https://orcid.org/0000-0002-4419-2345
Zoé Joly-Lopez ⓘ https://orcid.org/0000-0002-7926-322X
Michael D Purugganan ⓘ https://orcid.org/0000-0002-9197-4112

Reviewer #2 (Public review): https://doi.org/10.7554/eLife.99352.3.sa1
Reviewer #3 (Public review): https://doi.org/10.7554/eLife.99352.3.sa2
Reviewer #4 (Public review): https://doi.org/10.7554/eLife.99352.3.sa3
Reviewer #5 (Public review): https://doi.org/10.7554/eLife.99352.3.sa4
Author response https://doi.org/10.7554/eLife.99352.3.sa5

## Additional files

### Supplementary files

MDAR checklist

Supplementary file 1. Systems genetics analysis of variance in the transcriptome of the Indica population in normal and salt conditions. Abbreviations: TrID - TranscriptID; DF - Degrees of Freedom; SS - Sum of Squares; MS - Mean Sum of Squares; F - F-statistic value; p - p-value; FDR - False Discovery rate; H2 - Broad-sense heritability; G - Genotype; E - Environment; GE - Genotype-by-Environment; R - Residuals.

Supplementary file 2. Summary statistic of selection on gene expression in the Indica population in normal and salt conditions. Abbreviations: P-value: Two-sided Mann-Whitney U-test; S: Linear selection differential; C: Quadratic selection differential; $\mu$ and $\sigma$ - mean- and variance-standardized selection differential.

Supplementary file 3. Selection statistics of transcripts in normal and salt conditions. Abreviations: TrID - TranscriptID; S - Linear Selection Differential; C -Quadratic Selection Differential; $\mu$ and $\sigma$ - mean- and variance-standardized selection differential; tval - t-statistic associated with the estimation of seletion differentials; Pval - Pvalue associated with the estimation of selecion differentials; BonferroniPval - Bonferroni corrected Pvalue.

Supplementary file 4. Conditionally neutral, Antagonistic Pleiotropic (CANP) transcripts and their enrichments. Abbreviations: AP : Antagonistically Pleiotropic; FDR - False Discovery Rate.

Supplementary file 5. Biological Pathways and Processes under selection. Abbreviations: TrID - Transcript ID; Gene_id - Gene ID; GO_ID: Biology Gene Ontology (ID); S - Linear selection differential (raw) in the normal and saline field.

Supplementary file 6. PCA analyses of transcripts in the normal and saline conditions with PCs explaining at least 0.5% variation in total gene expression.

Supplementary file 7. Selection statistics of transcript PCs and its enrichment under normal and saline conditions. KEGG Pathway Enrichment of the PCs with linear significant selection gradeient in either normal or saline conditions.

Supplementary file 8. Evidence for salinity stress induced decoherence. Unique transcripts along with its frequency with significantly different correlation between environments.

Supplementary file 9. Linear and Quadratic Selection gradients and their associated statistics acting on the traits. Abbreviations: Beta-Linear gradient; Gamma-Quadratic gradient; SE-Standard error; df-Degree of Freedom; tvalue and Pval - t-statistic value and P value associated with estimation of beta and gamma.

Supplementary file 10. eQTLs in normal and saline environment. leadSNP trans-eQTLs with FDR < 0.001 in the saline conditions. Abbreviations: SNPid-SNP ID; Chr-Chromosome SNP is persent on; Pos- Position on chromosome for the SNP; Gene-Gene regulated by SNP; Statistic and Pvalue-Statistic and P value associaited with association analysis; FDR- False Discovery Rate; beta - Effect size of gene regulation by the SNP.

Supplementary file 11. trans-eQTL hotspots idetified in the saline environment. Abbreviations: HotspotID- Hotspot ID; Chr - Chromosome on which Hoptspot is present; StartPos and StopPos -

Start and End chromosomal location of the hopspot; BinWidth-Size (in bp) of the hotspot; Nogenes_ Salt- Number of genes regulated by SNPs in the hotspot region; RegulatedGeneID - ID of genes regulated by SNPs in the hotspot region.

Supplementary file 12. cis-trans compesating vs reinforcing transcripts identified in normal and saline conditions. Abbreviations: TrID - TranscriptID; mean cis-effect - mean effect of cis-regulatory variants; mean trans-effect - mean effect of trans-acting variants; Inter-varietal variation - inter-varietal variation in gene expression for the population; Group - Compensating/Reinforcing.

Supplementary file 13. Information regarding the accessions used in this study. Abbreviations: IRGC_ ID_DNA_Source: IRGC ID of the accesion; Varietal Name: Common varietal name of the accession; Varietal Group: Subgroup the accession belongs to; SRA Bioproject ID: SRA ID where the genomic data for each accession is available.

## Data availability

Raw sequence reads have been submitted to the NCBI BioProject under PRJNA1010833 and PRJNA1012700. Further the metadata along with the processed RNA expression counts can be found at Zenodo. The custom codes used for the analyses in the manuscript can be found at GitHub (copy archived at *Gupta, 2025*).

The following datasets were generated:

| Author(s) | Year | Dataset title | Dataset URL | Database and Identifier |
|---|---|---|---|---|
| Gupta S, Groen SC, Zaidem ML, Sajise AGC, Calic I, Natividad MA, McNally KL, Vergara GV, Satija R, Franks SJ, Singh RK, Joly-Lopez Z, Purugganan MD | 2024 | Genetic architecture of gene regulation under salinity stress in rice | https://www.ncbi.nlm. nih.gov/bioproject/ PRJNA1010833 | NCBI BioProject, PRJNA1010833 |
| Purugganan MD, Gupta S, Groen SC, Zaidem ML, Sajise AGC, Calic I, Natividad M, McNally KL, Vergara GV, Satija R, Franks SJ, Singh RK, Joly-Lopez Z, Purugganan MD | 2024 | System genomics of rice salinity stress | https://www.ncbi.nlm. nih.gov/bioproject/ PRJNA1012700 | NCBI BioProject, PRJNA1012700 |
| Gupta S, Groen SC, Zaidem ML, Sajise AG, Calic I, Natividad MA, McNally KL, Vergara GV, Franks SJ, Singh RK, Joly-Lopez Z, Purugganan MD | 2023 | Processed RNA expression count data and metadata from Gupta et al.:Systems genomics of salinity stress response in rice | https://doi.org/10. 5281/zenodo.8284531 | Zenodo, 10.5281/ zenodo.8284531 |

The following previously published datasets were used:

| Author(s) | Year | Dataset title | Dataset URL | Database and Identifier |
|---|---|---|---|---|
| 3,000 rice genomes project | 2014 | The 3000 Rice Genomes Project | https://www.ncbi.nlm. nih.gov/bioproject/ PRJEB6180 | NCBI BioProject, PRJEB6180 |

*Continued on next page*

*Continued*

| Author(s) | Year | Dataset title | Dataset URL | Database and Identifier |
|---|---|---|---|---|
| Groen SC, Calic I, Joly-Lopez Z, Platts AE, Choi JY, Natividad M, Dorph K, Mauck WM, Bracken B, Cabral CLU, Kumar A, Torres RO, Satija R, Vergara G, Henry A, Franks SJ, Purugganan MD | 2019 | Population genome sequencing of Asian rice Oryza sativa varities | https://www.ncbi.nlm.nih.gov/bioproject/PRJNA557122 | NCBI BioProject, PRJNA557122 |

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
