## [Editor Report · eLife Assessment]

Working with a diverse panel of rice accessions grown in field conditions, this **valuable** study measures changes in transcript abundance, tests for patterns of selection on gene expression, and maps the genetic basic of variation in gene expression in normal and elevated salinity treatments. The manuscript provides **solid** evidence that mean gene expression levels are further from the optimum abundance for more genes under the elevated salinity treatment compared to normal treatment, and that a relatively small number of genes are hotspots that harbor genetic variants which affect broader genome-wide patterns of natural variation in gene expression under high salinity conditions. However, the design, clarity, and interpretation of several statistical analyses can be improved, some opportunities for integration among datasets and analyses could yet be realized, and genetic manipulation is required to confirm functional involvement of any specific genes in regulatory networks or organismal traits that confer adaptation to higher salinity conditions. The manuscript will be of interest to evolutionary biologists studying the genetics of complex traits and a resource for plant biologists studying mechanisms of abiotic stress tolerance.

---

## [Referee Report · Reviewer #2 (Public review)]

The authors investigate the gene expression variation in a rice diversity panel under normal and saline growth conditions to gain insight into the underlying molecular adaptive response to salinity. They present a convincing case to demonstrate that environment stress can induce selective pressure on gene expression, which is in agreement with their earlier study (Groen et al, 2020). The data seems to be a good fit for their study and overall the analytic approach is robust.

(1) The work started by investigating the effect of genotype and their interaction at each transcript level using 3'-end-biased mRNA sequencing, and detect a wide-spread GXE effect. Later, using the total filled grain number as a proxy of fitness, they estimated the strength of selection on each transcript and reported stronger selective pressure in saline environment. However, this current framework rely on precise estimation of fitness and, therefore can be sensitive to the choice of fitness proxy.

(2) Furthermore, the authors decomposed the genetic architecture of expression variation into cis- and trans-eQTL in each environment separately and reported more unique environment specific trans-eQTLs than cis-. The relative contribution of cis- and trans-eQTL depends on both the abundance and effect size. I wonder why the latter was not reported while comparing these two different genetic architectures. If the authors were to compare the variation explained by these two categories of eQTL instead of their frequency, would the inference that trans-eQTLs are primarily associated with expression variation still hold?

(3) Next, the authors investigated the relationship between cis- and trans-eQTLs at transcript level and revealed an excess of reinforcement over compensation pattern. Here, I struggle to understand the motivation for testing the relationship by comparing the effect of cis-QTL with the mean effect of all trans-eQTLs of a given transcript. My concern is that taking the mean can diminish the effect of small trans-eQTLs potentially biasing the relationship towards the large-effect eQTLs.

Comments on latest version:

After the revision, the article has improved substantially. The authors have addressed most of my concerns and suggestions, except for testing the eQTL reinforcement/compensation relationship in the context of genetic architecture. I understand the motivation for testing this relationship at the gene level to determine whether it arises from directional or stabilizing selection, rather than examining it in a cis-trans pairwise fashion. However, I find the definition of this relationship unclear. The authors state in line 824 that "Genes were defined as compensating and reinforcing if they had at least 60% of individuals with opposite and same cis-trans allelic configuration, respectively." In contrast, if I understood correctly, the response to reviewers describes the relationship as reinforcing if the cis-eQTL effect is in the same direction as the mean effect of all the detected trans-eQTLs. I would request that the authors clarify their method of defining this relationship. Also, one should be aware of the fact that this relationship can evolve neutrally. Since there was no formal test performed to say it is otherwise, the authors might need to interpret the relationship carefully.

While the authors explain the possible factors that could lead to the trend of observing widespread genotype-dependent plastic responsse without significant genotype-dependent plasticity for fitness (L142), it is also important to consider the time axis. While filled grain serves as a proxy for fitness over time, gene expression profiles provide only a snapshot at a given time point. Therefore, temporal GxE dynamics may also play a role here.

Also, I am a little surprised by not mentioning anything about the code availability in this manuscript. I would request the authors to incorporate that in the revised version.

---

## [Referee Report · Reviewer #3 (Public review)]

In this work, the authors conducted a large-scale field trial of 130 indica accessions in normal vs. moderate salt stress conditions. The experiment consists of 3 replicates for each accession in each treatment, making it 780 plants in total. Leaf transcriptome, plant traits, and final yield were collected. Starting from a quantitative genetics framework, the authors first dissected the heritability and selection forces acting on gene expression. After summarizing the selection force acting on gene expression (or plant traits) in each environment, the authors described the difference in gene expression correlation between environments. The final part consists of eQTL investigation and categorizing cis- and trans-effects acting on gene expression.

Building on the group's previous study and using a similar methodology (Groen et al. 2020, 2021), the unique aspect of this study is in incorporating large-scale empirical field works and combining gene expression data with plant traits. Unlike many systems biology studies, this study strongly emphasizes the quantitative genetics perspective and investigates the empirical fitness effects of gene expression data. The large amounts of RNAseq data (one sample for each plant individual) also allow heritability calculation. This study also utilizes the population genetics perspective to test for traces of selection around eQTL. As there are too many genes to fit in multiple regression (for selection analysis) and to construct the G-matrix (for breeder's equation), grouping genes into PCs is a very good idea.

In the previous review, three major points were mentioned. The manuscript was modified, and here I briefly summarize them as a reference for future works:

(1) The separate sections (selection analysis, transcript correlation structure change, and eQTL) could use better integration.

(2) It would be worth considering joint analyses integrating the two environments together.

(3) Whether gene expression PCs or unique expression modules should be used in selection analyses.

Regarding whether to use PCs or WGCNA eigengenes to summarize gene expression for selection analyses, the authors reported that only a few WGCNA eigengenes were under selection, citing this observation as the rationale for choosing PC over eigengenes. However, as the relative false positive-negative rates of these choices likely require another dedicated study to explore, at this stage, it might be premature to state which method is better based on which gives more positive results. On one hand, one could easily imagine that plants screwed up by salinity have erratic genomewide expression and become extreme data points on the PCs, making the PCs a good proxy to correlate with fitness. On the other, it remains to be discussed whether this genomewide screwed-up-ness is what we want to measure in this study or whether we should focus on more dedicated gene modules instead. I suggest the authors acknowledge both possibilities. In this revision, I do not see relevant WGCNA results (as mentioned in the previous response letter) reported.

Figure 4: The observation that chlorophyll a content is under negative selection under BOTH conditions is a bit counterintuitive. The manuscript only mentioned "consistent with the general trend for reduced photosynthesis under salinity stress" (line 329) but did not mention why this increased fitness, even in normal conditions.

---

## [Referee Report · Reviewer #4 (Public review)]

The manuscript examines how patterns of selection on gene expression differ between a normal field environment and a field environment with elevated salinity based upon transcript abundances obtained from leaves of a diverse panel of rice germplasm. In addition, the manuscript also maps expression QTL (eQTL) that explains variation in each environment. One highlight from the mapping is that a small group of trans-mapping regulators explains some gene expression variation for large sets of transcripts in each environment.

The overall scope of the datasets is impressive, combining large field studies that capture information about fecundity, gene expression, and trait variation at multiple sites. The finding related to patterns indicating increased LD among eQTLs that have cis-trans compensatory or reinforcing effects in interesting in the context of other recent work finding patterns of epistatic selection. The authors have made some changes that address previous comments. However, some analyses in the manuscript remain less compelling or do not make the most from the value of collected data. Although the authors have made several improvements to the precision with which field-specific terminology is applied and to the language chosen when interpreting analytical findings, additional changes to improve these aspects of the manuscript remain necessary.

Selection of gene expression: One strength of the dataset is that gene expression and fecundity were measured for the same genotypes in multiple environments. However, the selection analyses are largely conducted within environments. Addition of phenotypic selection analyses that jointly analyze gene expression across environments and or selection on reaction norms would be worthwhile.

Gene expression trade-offs: The terminology and possibly methods involved in the section on gene expression trade-offs need amendment. I specifically recommend discontinuing reference to the analysis presented as an analysis of antagonistic pleiotropy (rather than more general as trade-offs) because pleiotropy is defined as a property of a genotype, not a phenotype. Gene expression levels are a molecular phenotype, influenced by both genotype and the environment. By conducting analyses of selection within environments as reported, the analysis does not account for the fact that the distribution of phenotypic values, the fitness surface, or both may differ across environments. Thus, this presents a very different situation than asking whether the genotypic effect of a QTL on fitness differs across environments, which is the context in which the contrasting terms antagonistic pleiotropy and conditional neutrality have been traditionally applied. The results reported do not persuasively support the assertion made in the response to reviewers that the terminology is reasonable due to strong coupling between genotype and phenotype. A more interesting analysis would be to examine whether the covariance of phenotype with fitness has truly changed between environments or whether the phenotypic distribution has just shifted to a different area of a static fitness surface.

Biological processes under selection / Decoherence: In the initial review, it was noted that PCA is likely not the most ideal way to cluster genes to generate consolidated metrics for a selection gradient analysis. Because individual genes will contribute to multiple PCs, the current fractional majority-rule method applied to determine whether a PC is under direct or indirect selection for increased or decreased expression comes across as arbitrary and with the potential for double-counting genes. A gene co-expression network analysis could be more appropriate, as genes only belong to one module and one can examine how selection is acting on the eigengene of a co-expression module. Building gene co-expression modules would also provide a complementary and more concrete framework for evaluating whether salinity stress induces "decoherence" and which functional groups of genes are most impacted. Although results of co-expression network analyses are now briefly discussed in the response to reviewers, the findings and their relationship to the PCA/"decoherence" analyses are not reported in the manuscript.

Selection of traits: Having paired organismal and molecular trait data is a strength of the manuscript, but the organismal trait data are underutilized. The manuscript as written only makes weak indirect inferences based on GO categories or assumed gene functions to connect selection at the organismal and molecular levels. After prompted by the initial reviews to test for correspondence between SNPs that explain organismal and gene expression trait variation or co-variance of co-expression module variation and trait variation, the response to reviewers indicates finding negative results. These findings should be included in the manuscript text and discussed.

Genetic architecture of gene expression variation: More descriptive statistics of the eQTL analysis have been included, although additional information about the variation in these measures within environments would be useful. The motivation for featuring patterns of cis-trans compensation specifically for the results obtained under high salinity conditions remains unclear to me. If the lines sampled have predominantly evolved under low salinity conditions, and the hypothesis being evaluated relates to historical experience of stabilizing selection, then evaluating the eQTL patterns under normal conditions provides the more relevant test of the hypothesis.

Lines 280-282: The revised sentence continues to read as an overstatement and merits additional revision with citations.

Lines 379-381: Following revision, it still remains unclear how the interpretation follows from the above analysis; the inference as written goes significantly beyond what may be specifically inferable from the result.

---

## [Referee Report · Reviewer #5 (Public review)]

Summary:

The researchers examined selection across multiple levels, including gene expression, biological processes, and regulatory mechanisms, with a particular focus on comparing selection between different environmental conditions. They further explored potential evolutionary mechanisms. This is made possible with a comprehensive dataset comprising gene expression data from 130 accessions with three replicates collected in two environments in the field, genomic data from 125 genotypes, and associated physiological traits. The findings have significant implications for understanding the evolution of stress adaptation, and the identified possible genes and pathways for further investigation.

The researchers began by focusing on the selection of gene expression across two environments, comparing the number of genes under selection and the effect sizes, as well as examining how selection in each environment acts on the same individual genes. They then expanded their analysis to consider selection in biological processes, investigating the relationships between selection acting on individual genes within processes and selection acting among different processes.

Additionally, they explored selection at the organismal level by examining traits.

The study further transitioned from analyzing individual gene expression to investigating gene-gene interactions. They briefly examined correlation variation among gene pairs between the two conditions, identifying pairs with rewired interactions that suggest potential selection on gene regulation or the effect of rewiring on tolerance. The researchers then delved into the genetic architecture underlying these patterns by mapping eQTLs. Their comparison of cis- and trans-eQTLs revealed that trans-eQTLs were more variable across conditions. Notably, they identified hotspots representing master regulators that possibly underlie the greater variability of trans-eQTLs across environments. They further discovered that trans-eQTLs are generally under purifying selection (particularly in salt conditions), while cis-eQTLs are under balancing selection, exhibiting higher nucleotide diversity. As for how cis- and trans-eQTL effects combine at the level of individual genes, more are found to be reinforced and the hypothesis of genetic fixation on cis- and trans-eQTL effects combination is further tested.

Strengths:

A key strength of this study is its comprehensive approach, extending beyond the analysis of gene expression to include gene-gene interactions, genetic architectures, and selections of genetic regulation factors. The exploration of gene expression selection through its connection with fitness, as introduced in the researchers' previous work, provides valuable insights into the role of gene expression in adaptation. The study investigates selection across multiple levels of biological responses, including individual gene expression, genes associated with biological processes, gene-gene interactions, and the underlying genetic architecture. The experimental design enables a direct comparison of selection between control and salinity conditions, which sheds light on the effects of stress on selection and the dynamics of adaptation to stress. Additionally, the manuscript is well-written, with a clear connection to current literature. The discussion effectively integrates findings with broader implications, making it a satisfying read.

Weaknesses:

The lack of formal testing for environment-specific selections (e.g., selection of gene expression specifically in salinity stress, PCs, or traits) is a major limitation, as previous reviewers have flagged. Explicit tests of eQTLs variation between conditions are introduced, so similar formal tests should also be introduced in selection sections. For example, a formal test of selections of gene expression might be helpful to solve variance/mean- standardization concerns between two environments.

Additionally, some aspects of the analysis appear somewhat arbitrary and could benefit from further sensitivity testing. Line 203: The concern about bias in detecting more CN than AP, as mentioned by the authors and previously flagged by reviewers, does not seem fully resolved with the current methods given the arbitrary cut-off. Incorporating additional tests suggesting the conclusion is insensitive to the cutoff would be very helpful. Similar is the classification of genes into compensatory and reinforcing categories based on 60% of individuals as a cutoff.

While this study focuses on gene regulation, its connection with the selection of gene expression and biological pathways is not well integrated. In particular, the discovery of eQTLs is not explicitly linked to gene expression selection or biological pathways, leaving this relationship underexplored. Suggestive comments: Currently the summarization of selection is based on eQTLs. It would be interesting to also summarize the selection patterns identified from previous sections based on genes being cis/trans-regulated. Moreover, it might be interesting to see if there is more loss or gain of eQTLs under salt stress and their functions. The current results mentioned variations of eQTLs but not clear if they are loss or gain. E.g., one way is to identify genes related to cis and trans-eQTLs and see their correlation changes with genes being regulated using CILP (also as a way to informatively narrow down gene pairs for CILP).

Similarly, the section on selection at the organismal trait level appears disconnected from the rest of the analysis (e.g., if it is not tested to be related to other features, mentioning why it might not be related would be helpful). Admittedly, the discussion of how biological processes discovered at different levels integrate together is helpful.

Other comments: given there is no comparison between loss of coherence (correlations) and gain of coherence under salinity stress to show the dominant role of decoherence, maybe need to also discuss the genes and processes related to the gain of coherence? This is because the understanding of activation (gain of coherence) of some regulations/processes under stress conditions could also be interesting. It is not clear if decoherence (e.g., lines 293-296) refers to significant correlation changes or just loss of the correlation in salinity stress.

---

## [Author Response]

The following is the authors’ response to the original reviews.

**Reviewer #1 (Public Review):**
Summary:Understanding the mechanisms of how organisms respond to environmental stresses is a key goal of biological research. Assessment of transcriptional responses to stress can provide some insights into those underlying mechanisms. The researchers quantified traits, fitness, and gene expression (transcriptional) response to salinity stress (control vs stress treatments) for 130 accessions of rice (three replicates for each accession), which were grown in the field in the Philippines. This experimental design allowed for many different types of downstream analyses to better understand the biology of the system. These analyses included estimating the strength of selection imposed on transcription in each environment, evaluating possible trade-offs in gene expression, testing whether salinity induces transcriptional decoherence, and conducting various eQTL-type analyses.Strengths:The study provides an extensive analysis of gene expression responses to stress in rice and offers some insights into underlying mechanisms of salinity responses in this important crop system. The fact that the study was conducted under field conditions is a major plus, as the gene expression responses to soil salinity are more realistic than if the study was conducted in a greenhouse or growth chamber. The preprint is generally well-written and the methods and results are mostly well-described.Weaknesses:While the study makes good use of analyzing the dataset, it is not clear how the current work advances our understanding of gene regulatory evolution or plant responses to soil salinity generally. Overall, the results are consistent with other prior studies of gene expression and studies of selection across environmental conditions. Some of the framing of the paper suggests that there is more novelty to this study than there is in reality. That said, the results will certainly be useful for those working in rice and should be interesting to scientists interested in how gene expression responses to stress occur under field conditions. I detail other concerns I had about the preprint below:The abstract on lines 33-35 illustrates some of my concerns about the overstatement of the novelty of the current study. For example, is it really true that the role of gene expression in mediating stress response and adaptation is largely unexplored? There have been numerous studies that have evaluated gene expression responses to stresses in a wide range of organisms. Perhaps, I am missing something critically different about this study. If so, I would recommend that the authors reword this sentence to clarify what gap is being filled by this study. Further, is it really the case that none of them have evaluated how the correlational structure of gene expression changes in response to stresses in plants, as implied in lines 263-265? Don't the various modules and PC analyses of gene expression get at this question?

We have re-worded these sentences, and highlighted the novelty of our work.

There were some places in the methods of the preprint that required more information to properly evaluate. For example, more information should be provided on lines 664-668 about how G, E, and GxE effects were established, especially since this is so central to this study. What programs/software (R? SAS? Other?) were used for these analyses? If R, how were the ANOVAs/models fit? What type of ANOVA was used? How exactly was significance determined for each term? Which effects were considered fixed and which were random? If the goal was to fit mixed models, why not use an approach like voom-limma (Law et al. 2014 Genome Biology)? More details should also be added to lines 688-709 about these analyses, including what software/programs were used for these analyses.

We have added more details in the methods. Also, although we could in priciple use voom-limma to fit our mixed model, to be able to partition variance into G, E and G×E, we need to use the function fitExtractVarPartModel (from package VariancePartition) which requires all categorical variables to be modeled as random effects. Therefore, we couldn’t model environment as a fixed effect.

One thing that I found a bit confusing throughout was the intermixing of different terms and types of selection. In particular, there seemed to be some inconsistencies with the usage of quantitative genetics terms for selection (e.g. directional, stabilizing) vs molecular evolution terms for selection (e.g. positive, purifying). I would encourage the authors to think carefully about what they mean by each of these terms and make sure that those definitions are consistently applied here.

We have defined the selection terms used in the study and used these terms consistently throughout the manuscript.

It would be useful to clarify the reasons for the inherent bias in the detection of conditional neutrality (CN) and antagonistic pleiotropy (AP; Lines 187-196). It is also not clear to me what the authors did to deal with the bias in terms of adjusting P-value thresholds for CN and AP the way it is currently written. Further, I found the discussion of antagonistic pleiotropy and conditional neutrality to be a bit confusing for a couple of reasons, especially around lines 489-491. First of all, does it really make sense to contrast gene expression versus local adaptation, when lots of local adaptation likely involves changes in gene expression? Second, the implication that antagonistic pleiotropy is more common for local adaptation than the results found in this study seems questionable. Conditional neutrality appears to be more common for local adaptation as well: see Table 2 of Wadgymar et al. 2017 Methods in Ecology and Evolution. That all said, it is always difficult to conclude that there are no trade-offs (antagonistic pleiotropy) for a particular locus, as the detecting trade-offs may only manifest in some years and not others and can require large sample sizes if they are subtle in effect.

We have now explained the cause of the inherent bias in the detection of CN, and also elaborated on how we deal with this bias. Also, we have edited our discussion and added relevant citations to indicate both conditional neutrality and antagonistic pleiotropy can lead to local adaptations and added the caveat regarding detecting antagonistic pleiotropy.

**Reviewer #2 (Public Review):**
The authors investigate the gene expression variation in a rice diversity panel under normal and saline growth conditions to gain insight into the underlying molecular adaptive response to salinity. They present a convincing case to demonstrate that environmental stress can induce selective pressure on gene expression, which is in agreement to their earlier study (Groen et al, 2020). The data seems to be a good fit for their study and overall the analytic approach is robust.(1) The work started by investigating the effect of genotype and their interaction at each transcript level using 3'-end-biased mRNA sequencing, and detecting a wide-spread GXE effect. Later, using the total filled grain number as a proxy of fitness, they estimated the strength of selection on each transcript and reported stronger selective pressure in a saline environment. However, this current framework relies on precise estimation of fitness and, therefore can be sensitive to the choice of fitness proxy.

We now acknowledge this caveat in the discussion.

(2) Furthermore, the authors decomposed the genetic architecture of expression variation into cis- and trans-eQTL in each environment separately and reported more unique environment-specific trans-eQTLs than cis-. The relative contribution of cis- and trans-eQTL depends on both the abundance and effect size. I wonder why the latter was not reported while comparing these two different genetic architectures. If the authors were to compare the variation explained by these two categories of eQTL instead of their frequency, would the inference that trans-eQTLs are primarily associated with expression variation still hold?

We have now also reported the effect sizes for both cis- and trans-eQTLs in the two environments and showed that the trans-eQTLs have higher effect sizes as compared to cis-eQTLs, indicating that they are able to explain higher proportion of variation in transcript abundances in the two environments.

(3) Next, the authors investigated the relationship between cis- and trans-eQTLs at the transcript level and revealed an excess of reinforcement over the compensation pattern. Here, I struggle to understand the motivation for testing the relationship by comparing the effect of cis-QTL with the mean effect of all trans-eQTLs of a given transcript. My concern is that taking the mean can diminish the effect of small trans-eQTLs potentially biasing the relationship towards the large-effect eQTLs.

We wanted to estimate compensating vs reinforcing effects, which essentially entails identifying genes that have opposing directionality of cis and trans-effects. To get the total trans-effect we decided to take the mean effect of trans-eQTLs. This mean was only used to identify the compensating/reinforcing genes and although the mean effects diminishes the effect of small trans-eQTLs, this mean was not used in downstream analyses.

**Reviewer #3 (Public Review):**
In this work, the authors conducted a large-scale field trial of 130 indica accessions in normal vs. moderate salt stress conditions. The experiment consists of 3 replicates for each accession in each treatment, making it 780 plants in total. Leaf transcriptome, plant traits, and final yield were collected. Starting from a quantitative genetics framework, the authors first dissected the heritability and selection forces acting on gene expression. After summarizing the selection force acting on gene expression (or plant traits) in each environment, the authors described the difference in gene expression correlation between environments. The final part consists of eQTL investigation and categorizing cis- and trans-effects acting on gene expression.Building on the group's previous study and using a similar methodology (Groen et al. 2020, 2021), the unique aspect of this study is in incorporating large-scale empirical field works and combining gene expression data with plant traits. Unlike many systems biology studies, this study strongly emphasizes the quantitative genetics perspective and investigates the empirical fitness effects of gene expression data. The large amounts of RNAseq data (one sample for each plant individual) also allow heritability calculation. This study also utilizes the population genetics perspective to test for traces of selection around eQTL. As there are too many genes to fit in multiple regression (for selection analysis) and to construct the G-matrix (for breeder's equation), grouping genes into PCs is a very good idea.Building on large amounts of data, this study conducted many analyses and described some patterns, but a central message or hypothesis would still be necessary. Currently, the selection analysis, transcript correlation structure change, and eQTL parts seem to be independent. The manuscript currently looks like a combination of several parallel works, and this is reflected in the Results, where each part has its own short introduction (e.g., 185-187, 261-266, 349-353). It would be great to discuss how these patterns observed could be translated to larger biological insights. On a related note, since this and the previous studies (focusing on dry-wet environments) use a similar methodology, one would also wonder what the conclusions from these studies would be. How do they agree or disagree with each other?

We acknowledge that the manuscript currently presents some analyses in a somewhat independent manner. Although it would be ideal to have a central hypothesis/message, our study is meant to broadly outline the various responses and fitness effects of salinity stress in rice. Throughout the manuscript, we have also included comparisons between our findings and that of our previous studies on drought stress to highlight any consistent themes or novel insights.

Many analyses were done separately for each environment, and results from these two environments are listed together for comparison. Especially for the eQTL part, no specific comparison was discussed between the two environments. It would be interesting to consider whether one could fit the data in more coherent models specifically modeling the X-by-environment effects, where X might be transcripts, PCs, traits, transcript-transcript correlation, or eQTLs.

We do plan to consider fitting models that explicitly incorporate X-by-environment interactions to provide a more detailed understanding of the genetics of plasticity between the two environments, but it is beyond the scope of this paper. This will be explored in a separate report.

As stated, grouping genes into PCs is a good idea, but although in theory, the PCs are orthogonal, each gene still has some loadings on each PC (ie. each PC is not controlled by a completely different set of genes). Another possibility is to use any gene grouping method, such as WGCNA, to group genes into modules and use the PC1 of each module. There, each module would consist of completely different sets of genes, and one would be more likely to separate the biological functions of each module. I wonder whether the authors could discuss the pros and cons of these methods.

We recognize that individual genes can contribute to multiple PCs, and this is precisely why we choose PCA clustering over WGCNA where one gene can belong to only one module. Our aim was to recognize all biological processes that could be under selection in either environment, and since one gene can be involved in various different processes, we wanted to identify the contribution of these genes to different processes which can be done effectively by a PCA analyses.

**Reviewer #4 (Public Review):**
The manuscript examines how patterns of selection on gene expression differ between a normal field environment and a field environment with elevated salinity based on transcript abundances obtained from leaves of a diverse panel of rice germplasm. In addition, the manuscript also maps expression QTL (eQTL) that explains variation in each environment. One highlight from the mapping is that a small group of trans-mapping regulators explains some gene expression variation for large sets of transcripts in each environment. The overall scope of the datasets is impressive, combining large field studies that capture information about fecundity, gene expression, and trait variation at multiple sites. The finding related to patterns indicating increased LD among eQTLs that have cis-trans compensatory or reinforcing effects is interesting in the context of other recent work finding patterns of epistatic selection. However, other analyses in the manuscript are less compelling or do not make the most of the value of collected data. Revisions are also warranted to improve the precision with which field-specific terminology is applied and the language chosen when interpreting analytical findings.Selection of gene expression:One strength of the dataset is that gene expression and fecundity were measured for the same genotypes in multiple environments. However, the selection analyses are largely conducted within environments. The addition of phenotypic selection analyses that jointly analyze gene expression across environments and or selection on reaction norms would be worthwhile.

We do plan to consider fitting models that explicitly incorporate G×E interactions to provide a more detailed understanding of the genetics of plasticity between the two environments, but it is beyond the scope of this paper. This will be explored in a separate report.

Gene expression trade-offs:The terminology and possibly methods involved in the section on gene expression trade-offs need amendment. I specifically recommend discontinuing reference to the analysis presented as an analysis of antagonistic pleiotropy (rather than more general trade-offs) because pleiotropy is defined as a property of a genotype, not a phenotype. Gene expression levels are a molecular phenotype, influenced by both genotype and the environment. By conducting analyses of selection within environments as reported, the analysis does not account for the fact that the distribution of phenotypic values, the fitness surface, or both may differ across environments. Thus, this presents a very different situation than asking whether the genotypic effect of a QTL on fitness differs across environments, which is the context in which the contrasting terms antagonistic pleiotropy and conditional neutrality have been traditionally applied. A more interesting analysis would be to examine whether the covariance of phenotype with fitness has truly changed between environments or whether the phenotypic distribution has just shifted to a different area of a static fitness surface.

We recognize that pleiotropy is a property of a genotype, and not phenotype, but since our phenotype (gene expression) is strongly coupled with the genotype, we choose to call trade-offs as antagonistic pleiotropy. That being said, we did test whether the covariance of gene expression with phenotype significantly varies between environments, and found that to indeed be the case.

Biological processes under selection / Decoherence: PCs are likely not the most ideal way to cluster genes to generate consolidated metrics for a selection gradient analysis. Because individual genes will contribute to multiple PCs, the current fractional majority-rule method applied to determine whether a PC is under direct or indirect selection for increased or decreased expression comes across as arbitrary and with the potential for double-counting genes. A gene co-expression network analysis could be more appropriate, as genes only belong to one module and one can examine how selection is acting on the eigengene of a co-expression module. Building gene co-expression modules would also provide a complementary and more concrete framework for evaluating whether salinity stress induces "decoherence" and which functional groups of genes are most impacted.

We recognize that individual genes can contribute to multiple PCs, and this is precisely why we choose PCA clustering over WGCNA where one gene can belong to only one module. Our aim was to recognize all biological processes that could be under selection in either environment, and since one gene can be involved in various different processes, we wanted to identify the contribution of these genes to different processes which can be done effectively by a PCA analyses. But again as pointed out by the reviewer, our PCs did contain contribution (even negligible) of each gene, so to identify the ‘primary’ biological processes represented by the PCs, we chose the majority rule. As for testing decoherence, we agree that a co-expression module analyses would have provided additional support to the specific test performed in our manuscript, but since it would just be additional support, we choose to not add it in the manuscript.

But based on the recommendation of the reviewer(s), we did perform a WGCNA analyses and found a total of 14 and 13 modules in normal and saline conditions, of which 0 and 2 modules (with no significant GO enrichment) were under directional selection. This supports our reasoning of potentially missing on identification of processes under selection.

Selection of traits:Having paired organismal and molecular trait data is a strength of the manuscript, but the organismal trait data are underutilized. The manuscript as written only makes weak indirect inferences based on GO categories or assumed gene functions to connect selection at the organismal and molecular levels. Stronger connections could be made for instance by showing a selection of co-expression module eigengene values that are also correlated with traits that show similar patterns of selection, or by demonstrating that GWAS hits for trait variation co-localize to cis-mapping eQTL.

We did perform a GWAS for all the traits collected in both normal and saline environment, and only found significant hits for fecundity (in both normal and saline environment) and chlorophyll_a content (in the saline environment). But these regions did not overlap with any candidate genes or cis-mapping eQTL. Hence we choose to mention it in the manuscript. Additionally, using the WGCNA modules, we found that the only two module under selection in the saline environment were not significantly correlated with any of the traits measured.

Genetic architecture of gene expression variation:The descriptive statistics of the eQTL analysis summarize counts of eQTLs observed in each environment, but these numbers are not broken down to the molecular trait level (e.g., what are the median and range of cis- and trans-eQTLs per gene). In addition, genetic architecture is a combination of the numbers and relative effect sizes of the QTLs. It would be useful to provide information about the relative distributions of phenotypic variance explained by the cis- vs. trans- eQTLs and whether those distributions vary by environment. The motivation for examining patterns of cis-trans compensation specifically for the results obtained under high salinity conditions is unclear to me. If the lines sampled have predominantly evolved under low salinity conditions and the hypothesis being evaluated relates to historical experience of stabilizing selection, then my intuition is that evaluating the eQTL patterns under normal conditions provides the more relevant test of the hypothesis.

We have added the median number of eQTLs per gene in each environment. Additionally, we recognize that genetic architecture is a combination id numbers and effect size, and we have added information regarding the effect sizes of eQTLs by type and by environment as recommend by another reviewer. We did explore the distributions of phenotypic variance explained by the cis- vs. trans- eQTLs as recommended here, and found that trans-eQTLs explain more phenotypic variance than cis-eQTLs in both environments and that the distribution of either type of eQTL does not vary by environment. We are choosing to not add this in the main text due to space limitations. Lastly, we examined the patterns of cis-trans compensation/reinforcement under both normal and salinity conditions and have compared and contrasted the results from both in the main text.

**Recommendations for the authors:**

**Reviewer #1 (Recommendations For The Authors):**
Lines 126: I would recommend citing those who originally developed the 3' end targeted RNA sequencing methods (e.g. Meyer et al 2011 Molecular Ecology).

We have cited the recommended paper.

Lines 128-130: It would be useful to include a description here of what models were fit to the data to partition out G, E, and GxE effects.

Due to space limitations, we have in brief added a sentence to this effect.

Line 139: I would suggest changing "found little" to "no" since the test was not significant.

The sentence has been modified to say no evidence.

Line 313: I think you mean directional selection instead of positive selection.

We have corrected the text

Lines 362-363: Would the authors also expect an enrichment of reinforcing genes for most scenarios where that has been divergent selection, such as local adaptation among populations?

Based on our hypothesis, we would indeed expect an enrichment of reinforcing genes for scenarios of local adaptation where different alleles are maintained in different populations due to local adaptation.

**Reviewer #3 (Recommendations For The Authors):**
Figures 1d-e are not mentioned in the Results.

The figures have been referenced in appropriate places.

Lines 41-45: Terms such as reinforcement and compensation need to be explained in this specific context. Also "different selection regimes" is a bit broad and vague.

Due to word-count limitation, we are choosing to not elaborate the terms reinforcement and compensation in the abstract (since these are commonly used in the literature, and we have also defined these in the main text). Additionally, we now explicitly state the selection pressures associated with cis and trans eQTLs.

Table 1: Please explain S and C in the footnote.

We have added the recommended footnote

Figures: Some panel labels (a, b, c...) are mingled with the graphs.

We are re-made our figure such that the panel labels do not mingle with the plots.

Lines 588-591: font.

Modified

Lines 620-633: Please describe how these RNAseq libraries were allocated/pooled into different sequencing lanes to avoid potential batch effects among sequencing lanes.

The sequencing was performed on the same Illumina NextSeq 500 machine and we have added the sequencing libraries/pool plan in the methods (lines 688-689).

Lines 690-692: At the beginning of this paragraph, it was mentioned that the un-standardized coefficients were estimated. But here, it seems like the transcript data were already standardized in the data preparation step. What do lines 687-688 refer to? Further standardizing those estimated coefficients so that the whole distribution has mean=0 and sd=1?

Thank you pointing out our oversight. We checked our scripts and data preparation did not include transcript standardization, and we have removed the above line from the manuscript.

Lines 705-711: Please explain why assigning the positive/negative selection status for each gene is important. "Positive selection" here is defined as genes whose increased expression also increases fitness, but traditionally positive selection was defined as "the derived state is favored over the ancestral state". For a gene whose ancestral expression is high but lower expression increases fitness in this experiment, could we also say this gene is under positive selection? Given that we don't know the ancestral state here, maybe the authors could explain whether this definition is necessary. Also, given that many genes positively or negatively regulate each other in a pathway, it is also unclear whether it is necessary to assign the positive/negative status for a PC using the majority rule (lines 710-711).

We have now defined the different selection terms with respect to our study and use them consistently throughout the manuscript.

Lines 711-715: If I understand correctly, PCs were used as traits, and by definition PCs should all be orthogonal. Is this section saying only retaining PCs whose correlation < 0.6 with each other? What is the rationale?

PCA were performed on transcript abundance and the resulting orthogonal PCs explaining over 0.5% variance were all retained for selection analyses.

We also performed selection analyses on the functional traits measured in the field, but since these functional traits are correlated (and as such would not satisfy the independent variable requirement of regression analyses), we retained only those functional traits which had a Pearson correlation coefficient < 0.6.

Line 729: Please briefly describe what CLIP is doing.

We have added the required description.

Lines 736-741: The accession numbers do not add up to 125.

Thank you for catching our oversight. We have edited the text, and now the numbers add upto 125.

Line 796: Please remind readers where these 247k SNPs come from. Supposedly all accessions have been whole-genome sequenced, so the total number of SNPs should be larger than this.

We have detailed method detailing how the SNPs were obtained and processed in the lines preceding this. Indeed the number of SNPs would have been much bigger, but the stringent cutoffs and linkage disequilibrium pruning reduced our dataset to about 247k SNPs.

Lines 154-160: This is a bit confusing. The authors first mentioned, for the raw selection differentials, the mean and variance differ between environments, meaning they are misleading (why?). The next sentence then says non-standardized selection differentials will be used.

The mean and variance for transcript abundances vary between the two environments. Because traits are usually measured in different scales, it is recommended to standardize trait values using variance or mean before estimating selection coefficients. Multiplying this variance (or mean) standardized selection differential with heritability gives the expected response to selection in standard deviation (or mean) units. But if the trait variance (or mean) varies between traits or environments, it leads to a conflation between the standardized selection differential and trait variance (or mean), which can be misleading. So to avoid this, and given that our traits (transcript abundance in this case) were all measured on the same scale, we chose to not standardize our trait values and estimated raw selection differentials.

Figure 1 c-e: Please explain how the horizontal axis values were obtained. Is it assuming these selection differentials have a normal distribution of mean=0 & sd=1?

Yes, horizontal axis represents theorical quantile for selection differential assuming they have a normal distribution with mean=0 and sd=1. This has been added to the figure legend.

Line 162-168: Please clarify this part. What does “general trend towards stronger positive compared to negative selection on gene expression” mean? Does it mean the whole distribution of S is significantly different from 0, the difference in the number of genes in the S>0 vs S<0 category, or the a-bit-higher median |S| in the S>0 vs S<0 category? If it is the last one, are the small differences biological meaningful (0.053 vs. 0.047 for control & 0.051 vs. 0.050 for salt conditions), given that the authors defined |S|<0.1 as neutral?

By “general trend towards stronger positive compared to negative selection on gene expression”, we mean that more transcripts were under positive directional selection as compared to negative directional selection. We have also clarified this in the text now.

Line 177-178: This sentence implies disruptive selection is more important than stabilizing selection in the saline environment, but the test was not significant (line 176).

Although there was no significant difference in the magnitude of stabilizing vs disruptive selection *within* the saline environment, the number of transcripts experiencing stronger disruptive selection in the saline condition was greater than the number of transcripts experiencing disruptive selection in the normal conditions. And so comparing between conditions, disruptive selection plays an important role in the saline conditions.

Line 188-190: How CN vs. AP was statistically defined was not mentioned in the Methods section.

We have added in the main text within the Results section.

Line 203-214: How do these results fit with the previous observations that almost all transcripts have significant heritability?

Although we do find that all but three transcripts have a have significant genetic effect (and thus have significant heritability), the median broad-sense heritability for 51 antagonistically pleiotropic genes is 0.23. Give that, we would only be able to detect SNPs regulating gene expression with high effect size since our sample size is n=130. Additionally, we used a very stringent criteria (FDR < 0.001) to define eQTLs. These two factors in combination could lead to us not being able to detect significant eQTLs for AP genes.

Line 246-250: Please explain why the current conclusion would be opposite from the previous study. Supposedly the PCA, G matrix, and breeder’s equation were done for each environment separately. It makes sense that the G matrix and response to selection could be different between saline and drought treatments, but for the control treatments in the two studies, do they still differ? Why? Also in Table S7, it would be nice to show the % variation explained by each PC.

Although both our studies had largely overlapping samples, about 20% samples were unique to each study. Additionally, although the site where the study was performed was the same across the two studies, we found significant temporal differences in gene expression due to micro-environmental differences. Both these factors can lead to changes in direct and indirect selection and its response, and we are examining these differences as part of a separate study. We also highlight these caveats in our discussion.

Information on percent explained by each PCs is given in Table S5.

Figure 2b: The vertical axis was labeled as “selection gradient”, but I think the responses to selection (D, I, T) have different units.

We have re-labeled the vertical axis as “selection”.

**Reviewer #4 (Recommendations For The Authors):**
The manuscript mixes terminology for selection from quantitative genetics with that from population genetics. This is problematic, and the adjectives positive and negative should be replaced as descriptors of selection by instead rewording, for example, positive directional selection as directional selection for higher transcript abundance.Lines 193-196: The phrasing here reads as if the selection is solely acting on the presence/absence of expression rather than on quantitative variation in expression. During revision, it would be worth considering including an analysis of genes that parses genes that show the presence/absence of variation of expression within or across environments separately from genes that are expressed to non-trivial levels in both environments.

We have modified the sentence in question now. Also, we pre-processed RNA-seq data to remove all transcripts with low expression signals (sigma signal < 20), and further retained only transcripts that had non-trivial expression in at least 10% of the population, which we believe represents presence/absence of variation of expression within or across environments.

Lines 216-231: Is this analysis solely for directional selection? Not clear since previous sections examined both directional and stabilizing selection.

Yes, we performed this analysis for only directional selection, and have clarified this in the text too.

Lines 224-226: The meaning of this sentence is unclear and should be written more concretely.

We have rephrased the sentence to be more clear.

Lines 232-241: The description of the scientific logic here could be read as implying that genes interacting in networks are the sole source of indirect selection. I recommend revising the language to indicate this cause is one of several potential causes.

We have reworded the sentence such that we indicate selection acting on interacting genes is just one of the causes of indirect selection.

The strength of the conclusions of the decoherence analysis should be evaluated in light of caveats with such analyses (see Cai and Des Marais New Phytologist 2023).

We have added the caveat with relevant citation in the manuscript.

Rename this section as "Selection on Organismal Traits", as the previous sections have also been investigating selection on traits, just molecular traits.

We have renamed the section as recommended

Lines 314-318: Rewrite for clarity. Most environments select for an optimal phenotype; it is just the case here that the phenotypic distribution in the high salinity environment overlaps with the optimum.

We have rephrased and clarified the statement.

Lines 343-345: Rephrase to "These results indicate that natural variation in gene regulation under..."

Rephrased.

Line 354: "most" reads as too strong a descriptor here if the majority is ~60%.

We have reworded the sentence to read “more than half”

Lines 359-361: It is unclear to me how this interpretation follows from the above analysis.

We have reworded the sentence so that the claim follows our analysis.

Line 372: Is the expectation here more specifically one of epistatic selection? Other processes could stochastically lead to the genetic fixation of compensatory/reinforcing variants, but I think only epistasis for fitness would cause the interesting patterns of LD observed.

The expectation here is that certain cis and trans variants only exists to compensate/reinforce, potentially through epistasis. We have clarified this in the text.

Line 405: Change "adaptive organismal responses of organisms" to "organismal responses." As written, the sentence reads as being about plasticity rather than evolutionary responses, which are by populations, not organisms. None of the analyses included the manuscript test specifically test for adaptive plasticity.

Rephrased.